# *AKRMap*: Adaptive Kernel Regression for Trustworthy Visualization of Cross-Modal Embeddings

Yilin Ye [1 2]   Junchao Huang [3]   Xingchen Zeng [1]   Jiazhi Xia [4]   Wei Zeng [1 2]

## Abstract

Cross-modal embeddings form the foundation for multi-modal models. However, visualization methods for interpreting cross-modal embeddings have been primarily confined to traditional dimensionality reduction (DR) techniques like PCA and t-SNE. These DR methods primarily focus on feature distributions within a single modality, whilst failing to incorporate metrics (e.g., CLIPScore) across multiple modalities. This paper introduces *AKRMap*, a new DR technique designed to visualize cross-modal embeddings metric with enhanced accuracy by learning kernel regression of the metric landscape in the projection space. Specifically, *AKRMap* constructs a supervised projection network guided by a post-projection kernel regression loss, and employs adaptive generalized kernels that can be jointly optimized with the projection. This approach enables *AKRMap* to efficiently generate visualizations that capture complex metric distributions, while also supporting interactive features such as zoom and overlay for deeper exploration. Quantitative experiments demonstrate that *AKRMap* outperforms existing DR methods in generating more accurate and trustworthy visualizations. We further showcase the effectiveness of *AKRMap* in visualizing and comparing cross-modal embeddings for text-to-image models. Code and demo are available at https://github.com/yilinye/AKRMap.

## 1. Introduction

Cross-modal embeddings play a fundamental role for multi-modal models, functioning as cross-modal encoders (Rombach et al., 2022), objective functions (Ramesh et al., 2022), or evaluation metrics (Hessel et al., 2021; Wu et al., 2023a) for tasks like text-to-image (T2I) generation (Peebles & Xie, 2023; Hu et al., 2025). To assess the alignment of multi-modal models, various evaluation metrics based on these embeddings have been introduced, such as CLIPScore (Hessel et al., 2021) and Human Preference Score (HPS) (Wu et al., 2023a;b; Zhang et al., 2024). Despite their utility, embedding-based evaluations are often difficult to interpret, as the metrics are typically reported as aggregated values, without providing insights into instance-level performance. This deficiency undermines the trustworthiness and transparency of the evaluation.

To address this limitation, dimensionality reduction (DR) visualization serves as a crucial tool, offering a way to reveal the landscape of cross-modal metrics and enabling a more comprehensive understanding of model performance. Recent studies (Liang et al., 2022; Wang et al., 2023b;c) have explored the use of DR methods for cross-modal embeddings. These efforts often leverage established techniques such as PCA (Li et al., 2018), UMAP (McInnes et al., 2018) and t-SNE (Van der Maaten & Hinton, 2008), as well as autoencoder-based approaches (Le et al., 2018; Elhamod & Karpatne, 2024). However, existing DR methods are primarily designed to depict feature distributions within a single modality. When applied to cross-modal metrics, these methods often produce dense neighborhoods where points with significantly different metric values are positioned close, leading to overlap and local occlusion (Figure 1(b)). Moreover, multi-modal models are typically evaluated on large-scale datasets containing millions of data points. This creates a need for rendering contour maps that can reveal continuous metric distributions. However, local neighboring points with a mix of high and low metric values may be misrepresented, as the contour map depicts only a single aggregated value, causing inaccurate contour mapping as illustrated in Figure 1(c).

To construct trustworthy visualizations for cross-modal embeddings, a key consideration is to enhance the accuracy

---

[1]The Hong Kong University of Science and Technology (Guangzhou) [2]The Hong Kong University of Science and Technology [3]The Chinese University of Hong Kong (Shenzhen) [4]Central South University. Correspondence to: Wei Zeng <weizeng@hkust-gz.edu.cn>.

*Proceedings of the $42^{nd}$ International Conference on Machine Learning*, Vancouver, Canada. PMLR 267, 2025. Copyright 2025 by the author(s).

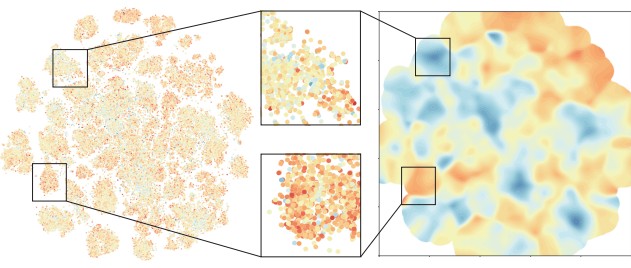

(a) t-SNE projection    (b) local occlusion    (c) inaccurate contour map

*Figure 1.* CLIPScore distribution on the COCO dataset by t-SNE (a). The visualization shows dense neighboring points with significantly different metric values, causing overlapping and occlusion (b) and highly inaccurate contour mapping (c).

of metric contour mapping in the projected 2D space. As contour estimation typically relies on radial basis function (RBF) kernel (e.g., Gaussian kernel), we seek to enhance the kernel-based mapping by coordinating it with the DR process. Drawing inspiration from supervised t-SNE (Hajderanj et al., 2019), we introduce Adaptive Kernel Regression Map (*AKRMap*), a supervised DR method that leverages adaptive kernel regression to effectively visualize the distribution of cross-modal embedding metrics. Specifically, *AKRMap* first constructs a supervised DR network explicitly guided by post-projection kernel regression loss with neighborhood constraint (Sect. 4.1.1). Next, to account for the gap between high-dimensional and low dimensional kernels, we improve the flexibility of the post-projection kernel through adaptive generalized kernel jointly learned with the projection (Sect. 4.1.2). This approach enables the generation of scatterplots for discrete data points and contour maps for continuous metric landscapes, while also supporting advanced features such as overlay views and zooming for interactive interpretation (Sect. 4.2). Both the DR method and the visualization tool have been implemented as a Python package and made publicly available.

We conduct quantitative experiments to evaluate *AKRMap* against traditional and autoencoder-based DR visualizations (Sect. 5.1). The results highlight the superior performance of *AKRMap* in accurately mapping cross-modal metrics for both in-sample and out-of-sample data points. We further demonstrate the practical applications of *AKRMap* across three distinct scenarios (Sect. 5.2): 1) visual exploration of human preference dataset (HPD) (Wu et al., 2023a), 2) visual comparison of diffusion-based and auto-regressive T2I models, and 3) visual examination of the global impact for fine-tuning. These applications illustrate that *AKRMap* effectively enables human-in-the-loop interpretation of model performance on large-scale datasets. Our method can also be potentially generalized to other modalities such as text-to-video task, with additional quantitative experiments shown in Appendix F.

In summary, our contributions are as follows:

- We propose a novel DR method for cross-modal metric visualization, which jointly learns the projection and metric contour mapping through kernel regression supervised DR with adaptive generalized kernel.

- We develop a tool for trustworthy visualization of cross-modal metrics, incorporating visualization features such as a scatterplot view and a contour map, along with interactive features like zooming and overlaying.

- We conduct quantitative experiments to demonstrate the superior performance of *AKRMap* in generating more accurate visualizations of cross-modal metric, and highlight its applications across three scenarios to enhance the trustworthiness of T2I model evaluation.

## 2. Related Work

### 2.1. Cross-Modal Embedding-based Evaluation

Cross-modal embeddings like CLIP (Radford et al., 2021) and ALIGN (Jia et al., 2021) form the foundation of multi-modal learning. Specifically, many evaluation methods for multi-modal models, such as T2I models, rely on these embeddings. Conventional metrics like Inception Score (Salimans et al., 2016) and Fréchet Inception Distance (Heusel et al., 2017) compute the average distance or distributional difference between generated and reference images in the embedding space, yet they are insufficient to measure cross-modal alignment and instance-level performance. With the advancement of multi-modal AI, cross-modal embedding metrics like CLIPScore (Hessel et al., 2021) have emerged to measure the alignment between prompts and generated images. While being effective in capturing the alignment, CLIPScore fails to model human preferences. To address this limitation, recent studies have proposed training specialized human preference models such as HPS (Wu et al., 2023a;b; Zhang et al., 2024), PickScore (Kirstain et al., 2023), and ImageReward (Xu et al., 2024), to better align with human judgments. For example, ImageReward (Xu et al., 2024) introduces a systematic pipeline for expert annotations to train preference models, while PickScore (Kirstain et al., 2023) leverages crowd-sourced human comparisons.

However, these metrics generally provide only an average score, offering a broad overview of a model's performance across the entire embedding space. This highlights the need for neural embedding visualizations that can expose the detailed distribution of metric values within the cross-modal embeddings, facilitating instance-level inspection and supporting human-in-the-loop evaluation.

### 2.2. Visualization for Neural Embeddings

Visualization is an important tool to support data analysis in conjunction with AI (Wang et al., 2023a; Ye et al., 2024).

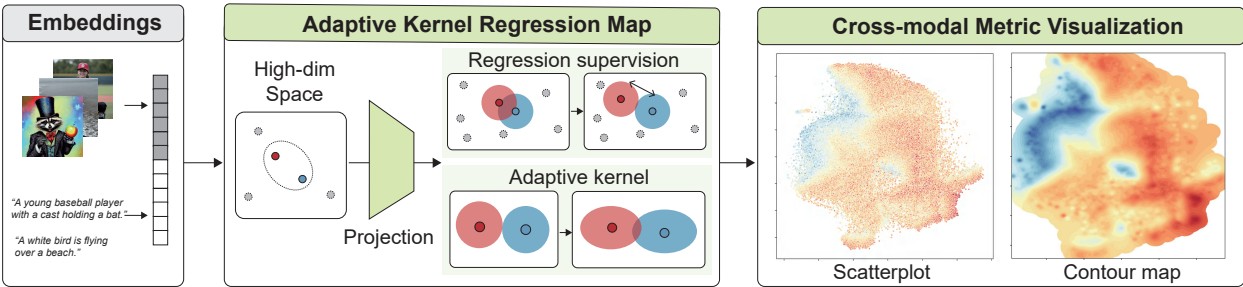

*Figure 2. AKRMap* is a neural network based DR method designed to learn adaptive kernel regression for visualizing cross-modal embeddings. The network integrates two key components to jointly learn data point projection and cross-modal metric estimation: 1) Kernel regression supervision, and 2) Adaptive generalized kernel. The resulting visualizations, including scatterplots and contour maps, provide a clearer and more accurate representation of the cross-modal metric distribution.

Particularly, it has proven to be effective for enhancing the understanding of various types of neural embeddings, including word embeddings (Mikolov et al., 2013; Chiang et al., 2020), vision-language embeddings (Liang et al., 2022; Wang et al., 2023b; Ye et al., 2025), and parameter spaces within loss landscape (Li et al., 2018; Elhamod & Karpatne, 2024). Commonly, dimensionality reduction (DR) techniques are employed for visualization. These include linear DR methods such as PCA (Abdi & Williams, 2010), as well as nonlinear DR methods like t-SNE (Van der Maaten & Hinton, 2008) and UMAP (McInnes et al., 2018). Parametric versions of traditional DR methods, which provide explicit mapping for projections, can be achieved by training neural networks, such as parametric t-SNE (Gisbrecht et al., 2015; Damrich et al., 2023), parametric UMAP (Sainburg et al., 2021) and other parametric methods for tasks like interactive clustering (Xia et al., 2023) or streaming data (Xia et al., 2024). Similarly, autoencoder-based DR methods (Le et al., 2018; Elhamod & Karpatne, 2024) have also been developed, incorporating an additional objective of reconstructing high-dimensional embeddings from their projections. Recently, there has been a growing interest in utilizing DR techniques to visualize cross-modal embeddings. For instance, UMAP has been employed to explore and visualize the modality gap between text and image embeddings (Liang et al., 2022; Wang et al., 2023b). To address this modality gap and enhance the cross-modal alignment in visualization space, some studies have introduced fusion-based DR methods (Ye et al., 2025), enabling the visualization of image embeddings in relation to text embedding anchor points.

However, previous efforts treat multi-modal embeddings separately, failing to effectively depict the landscape of cross-modal metrics (e.g., CLIPScore and HPS). In addition, existing metric-aware DR techniques focus on a limited range of particular metrics like distance (Sainburg et al., 2021) and density (Narayan et al., 2021), without a generalizable mechanism to preserve other different metrics. To address this limitation, we propose a kernel regression-based supervised projection method combined with an adaptive generalized kernel. Different from traditional kernel DR methods like kernel PCA (Schölkopf et al., 1997) that mainly use kernel to transform the high-dimensional features, we develop an adaptive kernel in the projection space to guide the DR. Notably, to construct the kernel guidance, we propose a novel cross-validation supervision technique that bridges between traditionally nonparametric kernel regression and parametric DR. This design allows the propagation of contour estimation errors back into the projection process. In this way, our method can dynamically adjust both the DR mapping and the kernel shape, enabling it to accurately fit the complex landscape of cross-modal metrics.

## 3. Problem Definition

Taking text-to-image generation models as an example, we formally define embedding-based metric for multi-modal models: A pretrained neural network $e(\cdot)$ encodes the prompt $t$ and the generated image $v$ into a high-dimensional embedding representations $e(t, v)$, which are subsequently used to predict a metric score $s = f(e)$. Here, $f(\cdot)$ represents the final operation applied to the embeddings, such as cosine distance in HPS (Wu et al., 2023a).

The problem of generating trustworthy visualizations for cross-modal embeddings can be defined as follows: given a dataset of prompt-image pairs $D = \{(t_i, v_i)\}_{i=1}^N$ and its embedding representations $e(D) = \{e(t_i, v_i)\}_{i=1}^N$, we first seek to learn a manifold $M$ (Goldberg et al., 2008) where $e(D)$ resides in the high-dimensional embedding space $R^d$. This manifold can be "spread out" (reparametrized) in the visualization space $R^2$ to show the distribution of metric score across the dataset, where the process can be modeled as a projection mapping $P(\cdot)$ $s.t.$ $P(e(D)) \in R^2$. This process aims to learn an explicit mapping function that projects high-dimensional embeddings to points in 2D space while accurately preserving the underlying metric distribution.

Next, to visualize a continuous distribution from the discrete projected sample points in $P(e(D))$, a contour map

in the 2D space needs to be estimated. Here, we adopt the approach used by contouring algorithms in various Python libraries, such as Plotly, by dividing the projected 2D space into a grid and calculating the metric distribution values at each grid point. Specifically, suppose the projected coordinates lie within a normalized 2D space of $[0, 1] \times [0, 1]$. For each grid $\mathbf{x}_g = (\frac{i}{N_w}, \frac{j}{N_h})$, we compute a value $\hat{s}(\mathbf{x}_g)$ based on the local metric distribution, which is then colored using a continuous colormap. $N_w$ and $N_h$ denote the number of grids along the x-axis and y-axis, respectively. In this manner, we can generate a contour map depicting continuous landscape of the metric distribution from the discrete projected sample points.

# 4. AKRMap

We propose Adaptive Kernel Regression Map (*AKRMap*), with the workflow illustrated in Figure 2. The input is a set of cross-modal embeddings with each high-dimensional vector representing a data point in the embedding space. First, to synchronize the contour mapping with the projection for accurate metric landscape estimation, we propose a cross-modal metric-supervised projection method comprising two key components: 1) Kernel Regression Supervision, which guides the projection to achieve more precise metric mapping (Sect. 4.1.1), and 2) an Adaptive Generalized Kernel, which accounts for the gap between high-dimensional and low-dimensional kernels and allows for more flexible contour mapping to capture complex metric landscapes (Sect. 4.1.2). Then, we design an interactive visualization tool to facilitate multi-scale exploration of metric distribution and individual data points (Sect. 4.2).

## 4.1. Adaptive Kernel Regression

### 4.1.1. KERNEL REGRESSION SUPERVISED PROJECTION

According to the problem definition, to achieve an accurate cross-modal metric visualization, cross-modal metric supervision is necessary. To address the challenge of constructing appropriate supervision for continuous cross-modal metric, we develop a kernel regression supervised projection method. Specifically, we propose learning a projection network $P: R^n \rightarrow R^2$ with a Nadaraya-Watson kernel regression (Ali et al., 2023) in the projected space:

$$\hat{s}(\mathbf{x}) = \frac{\sum_{k=1}^{N} K(\mathbf{x} - P(e(t_k, v_k))) \cdot s_k}{\sum_{k=1}^{N} K(\mathbf{x} - P(e(t_k, v_k)))}, \quad (1)$$

where $P(e(t_k, v_k)))$ is the projected sample point of metric embeddings and $s_k$ is the corresponding ground-truth metric value in the training set. $K(\cdot)$ is an RBF kernel in $R^2$.

Subsequently, to construct a supervised learning objective for the projection, inspired by the cross-validation method for kernel learning (Silverman, 2018), we randomly split

the dataset $D$ into training set $D_{tr}$ and validation set $D_{vl}$ of ratio $9 : 1$ in each epoch. Then, in equation (1), we estimate the metric distribution only with points in training set $((t_k, v_k) \in D_{tr})$. Next, we seek to minimize the weighted mean square error loss:

$$MSE_p = \frac{1}{|D_p|} \sum_{x_i \in D_p} (\hat{s}(x_i) - s_i)^2, \ p \in \{vl, tr\}, \quad (2)$$

$$MSE_r = w_1 MSE_{vl} + w_2 MSE_{tr}, \quad (3)$$

where the overall regression loss $MSE_r$ is a weighted sum of the loss on validation set ($MSE_{vl}$) and that on training set ($MSE_{tr}$). We seek to balance these loss terms to ensure mapping accuracy at both in-sample and out-of-sample positions on the contour, and the weights $w_1 = 1$ and $w_2 = 0.3$ are set empirically. The construction of this weighted trainval loss may differ from common practice but is motivated by our deeper thinking on the problem of connecting projection with kernel regression, which is further explained as follows. Unlike neural network methods, kernel regression is nonparametric, where the parameter like bandwidth is traditionally either precomputed or optimized via cross-validation. Thus, we are facing an interesting problem of how to bridge between parametric projection and nonparametric kernel regression, motivating us to leave a validation set to learn kernel parameters jointly with the neural network. The validation loss improves the generalization of kernel regression for unseen positions. However, relying solely on the validation loss will decrease the local detail of the map. In fact, as shown by additional experiments in Figure 14 in Appendix E, this train-val scheme proves to be essential to ensure robust mapping.

**Neighborhood Preservation Constraint**. To encourage the projection to maintain some neighborhood information of the high-dimensional embeddings, we combine our regression loss with a constraint term from traditional dimensionality reduction. Specifically, we incorporate the KL-divergence loss from t-SNE:

$$KL = \sum_{i}^{n} \sum_{j,j\neq i}^{n} p_{ij} \ln \frac{p_{ij}}{q_{ij}}, \quad (4)$$

where $p_{ij}$ and $q_{ij}$ are the neighborhood distribution probabilities in high-dimensional and low-dimensional space respectively, as detailed in (Van der Maaten & Hinton, 2008). Specifically, we adopt a perplexity-free implementation (De Bodt et al., 2018; Crecchi et al., 2020) of the KL loss for our parametric projection network. Overall, the objective function of our method is:

$$L = \lambda MSE_r + KL, \quad (5)$$

where $\lambda = 0.125$ is empirically set to balance the loss.

### 4.1.2. ADAPTIVE GENERALIZED KERNEL

In this section we illustrate the design of kernel $K(\cdot)$ in the regression supervision in equation (1). The key challenge is that we need to consider the discrepancy between high-dimensional and low-dimensional kernels. First, to avoid unstable optimization caused by exponentials and make the regression more compatible with our low-dimensional t-SNE neighborhood constraint, we adopt a t-distribution-like kernel instead of a Gaussian kernel. However, for complex metric distribution landscapes, especially those in embedding-based generative model evaluation, a standard kernel may lack enough flexibility. Particularly, it is difficult to determine a proper decay rate of the kernel value in relation to the distance. To address the problem, we take inspiration from the observation in a previous study (Narayan et al., 2021) of an approximate power law relationship between the local radius of the low-dimensional space ($r_e$) and that of original high-dimensional space $r_o$: $r_e(x_i) \approx \alpha \left[ r_o \left( x_i^h \right) \right]^{\beta}$. We hypothesize that this transformation can also improve the adaptability of low-dimensional kernel. Therefore, we propose using an adaptive generalized t-distribution kernel:

$$K(\mathbf{x}, \alpha, \beta) = \left( 1 + \alpha \left\| \mathbf{x} \right\|^{2\beta} \right)^{-1}, \qquad (6)$$

where $\alpha$ and $\beta$ are learnable parameters that are jointly optimized with the projection. In this way, our method can dynamically change the shape of the kernel to accurately fit the cross-modal metric landscape and reduce the risk of overfitting or underfitting due to suboptimal decay rate.

### 4.2. Cross-modal Metric Visualization

On the basis of *AKRMap*, we further develop an interactive visualization tool that provides two distinct views for exploring cross-modal embedding metrics:

- **Scatterplot view**. The scatterplot view displays all individual data points within the embedding space, with each point color-coded according to the cross-modal embedding metric. This view allows for direct interaction with the data points. Unlike scatterplots generated by baseline DR methods, which often suffer from significant occlusion that obscures the true metric distribution, *AKRMap* effectively reveals extreme values with greater clarity, as demonstrated in Figure 3.

- **Contour map**. For large datasets like HPD and COCO, scatterplot can become overcrowded with cluttered points. To address this limitation, we introduce a contour map that effectively represents cross-modal metric distribution in a continuous manner, by dividing the 2D space into grids and computing grid values based on the local metric distribution. This representation makes regions with extreme values more prominent and reveals distribution patterns with greater clarity.

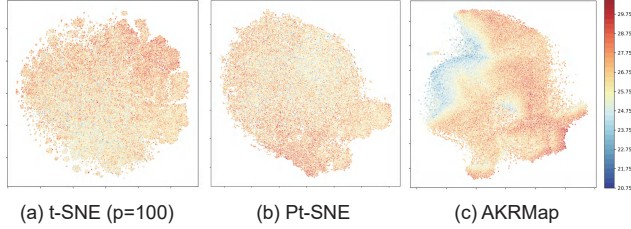

(a) t-SNE (p=100)    (b) Pt-SNE    (c) AKRMap

*Figure 3.* Comparison of scatterplots generated by t-SNE and *AKRMap* for the HPSv2 metric. Despite the visual clutter introduced by the large-scale dataset, *AKRMap* provides a clearer and more accurate representation of the HPSv2 metric distribution.

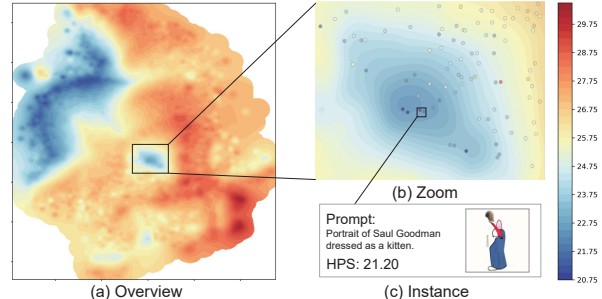

(a) Overview    (b) Zoom    (c) Instance

*Figure 4.* Contour map combined with zoom and overlay with point sampling for multiscale exploration of the HPD dataset.

The visualization tool includes various interactive features designed to facilitate in-depth exploration, including:

- **Zoom**. *AKRMap* enables efficient multiscale zooming by dynamically computing the contour map using varying grid resolutions, as illustrated in Figure 4. Specifically, it adjusts the level of detail by increasing the grid resolution proportionally to the zoom level, revealing finer details in the map as users zoom in. This capability is unique to *AKRMap* and not achievable with traditional DR methods, which lack the ability to preserve and display local details.

- **Overlay**. The scatterplot view and contour map allow data points to be overlaid onto the contour map, enabling users to simultaneously observe the overall distribution while interacting with individual data points, as shown in Figure 4. To mitigate occlusion, we have implemented sampling techniques, such as random sampling and Poisson disk sampling.

The visualization tool has been built into an easy-to-use python package with minimal dependencies, requiring only PyTorch and Plotly, which can be seamlessly integrated into interactive computational notebooks.

## 5. Experiment

### 5.1. Quantitative Experiments

**Baseline methods**. We compare *AKRMap* against three commonly used DR methods: PCA, t-SNE and UMAP, as

*Table 1.* Quantitative comparison of cross-modal embedding metric visualizations for test (out-of-sample) points on the HPD dataset. Our *AKRMap* outperforms baseline methods and ablations, achieving the best performance across all four cross-modal embedding metrics.

| Method | CLIPScore | | | HPSv2 | | | PickScore | | | Aesthetic Score | | |
|---|---|---|---|---|---|---|---|---|---|---|---|---|
| | $mae$ | $mape$ | $rmse$ | $mae$ | $mape$ | $rmse$ | $mae$ | $mape$ | $rmse$ | $mae$ | $mape$ | $rmse$ |
| PCA | 4.0042 | 16.5506 | 5.1142 | 1.4444 | 5.7870 | 1.8749 | 1.1497 | 5.9206 | 1.4686 | 0.5322 | 11.5645 | 0.6756 |
| t-SNE | 4.0241 | 16.6643 | 5.1523 | 1.4361 | 5.7568 | 1.8629 | 1.1579 | 5.9429 | 1.4640 | 0.4725 | 10.3128 | 0.6018 |
| UMAP | 4.0819 | 16.9066 | 5.2179 | 1.4432 | 5.7817 | 1.8725 | 1.2214 | 6.2721 | 1.5536 | 0.4463 | 9.6323 | 0.5721 |
| SAE | 4.1843 | 16.9386 | 5.2844 | 1.4483 | 5.8117 | 1.8696 | 1.2357 | 6.3952 | 1.5650 | 0.5809 | 12.7041 | 0.7251 |
| Neuro-Visualizer | 12.4201 | 43.8219 | 13.7318 | 1.6871 | 6.4526 | 2.0649 | 1.9655 | 9.5295 | 2.3427 | 0.4900 | 10.6306 | 0.6252 |
| AKRMap (w/o KR) | 4.0164 | 16.4830 | 5.1274 | 1.3978 | 5.6039 | 1.8102 | 1.1284 | 5.8004 | 1.4353 | 0.4752 | 10.3943 | 0.6003 |
| AKRMap (w/o GK) | 2.2812 | 9.1189 | 2.9458 | 1.1430 | 4.5498 | 1.4695 | 0.9064 | 4.6194 | 1.1686 | 0.4555 | 9.9267 | 0.5718 |
| AKRMap | **1.8707** | **7.3649** | **2.4253** | **0.8108** | **3.1935** | **1.1200** | **0.7712** | **3.9225** | **1.0225** | **0.4305** | **9.3340** | **0.5488** |

well as two encoder-based approaches: SAE (Le et al., 2018) and Neuro-Visualizer (Elhamod & Karpatne, 2024). For the encoder-based methods, the decoder is used to estimate grid colors, while the other baseline methods rely on the common Gaussian RBF kernel to estimate the 2D distribution map. Additional details about RBF parameter selection are provided in Appendix C. We use distance threshold to cut off empty areas for traditional DR techniques but keep the full landscape of autoencoder methods because far away positions in traditional DR results are ambiguous and can hardly be mapped back to original high-dimensional space without an inverse decoder. On the other hand, as autoencoder can easily lead to value explosion, we set an upper bound for the map to align with the maximum metric value in the dataset.

**Ablation Study**. We evaluate *AKRMap* under two alternative settings: 1) training the projection network using only the neighborhood constraint, without the supervision provided by Kernel Regression (AKRMap *w/o KR*), and 2) removing the adaptive Generalized Kernel component (AKRMap *w/o GR*).

**Dataset**. To evaluate performance on the cross-modal embedding metric, we select the widely used large-scale T2I dataset, the Human Preference Dataset (HPD) (Wu et al., 2023a). The HPD dataset contains over 430,000 unique images generated by various T2I models, along with their corresponding prompts, in the official training set, and 3,700 images in the test set.

**Cross-modal Embedding Metrics**. We compare *AKRMap* against baseline methods for visualizing several commonly used cross-modal embedding metrics in T2I generation, including CLIPScore (Hessel et al., 2021), HPSv2 (Wu et al., 2023a), PickScore (Kirstain et al., 2023), and a unimodal embedding metric commonly used in this cross-modal scenario, the Aesthetic Score (Schuhmann et al., 2022).

**Performance Evaluation**. We evaluate the accuracy and trustworthiness of the visualization methods using mapping errors, including $mae$, $rmse$, and $mape$. Specifically, we calculate these errors on the test set of HPD for out-of-sample points that were not used during training. We

also report the errors for in-sample points ($mae_{in}$, $mape_{in}$, $rmse_{in}$) from the HPD training set in Appendix B. This dual evaluation is important because a good visualization method must accurately represent the training data distribution while avoiding misleading maps caused by overfitting and ensuring reliable accuracy for test points.

**Training implementations**. The architecture of our projection network is a 4-layer MLP, with shape $(d, d)$ in the first three layers and $(d, 2)$ in the last layer, where $d$ is the dimensions of the input embeddings. Batch normalization and ReLU activation are applied to each layer. Our projection model is trained on one Nvidia L4 GPU with batch size of 1000 and 20 epochs. We use Adam optimizer with a learning rate of 0.002. For the t-SNE and PCA implementations, we use the python sklearn package, where the t-SNE method adopts the Barnes-Hut t-SNE (Van Der Maaten, 2014). For the UMAP implementation, we adopt the python umap-learn package. The Neuro-Visualizer implementation is based on the open-sourced code of the original paper (Elhamod & Karpatne, 2024), while the SAE is based on a github reimplementation[1].

The experiment results are presented in Table 1.

**Comparison with baselines**. Our method *AKRMap* consistently outperforms baseline methods in mapping accuracy across all the embedding metrics. Notably, *AKRMap* effectively reduces mapping errors for both training (in-sample) points (see Appendix B) and test (out-of-sample) points, demonstrating its ability to produce more trustworthy mappings. For instance, when applied to the HPSv2 metric, *AKRMap* reduces the MAE by nearly 50% for in-sample points and approximately 43% for out-of-sample points compared to the best-performing baseline, t-SNE. In addition, while autoencoder methods are effective for loss landscape, it has shown weakness in cross-modal metric mapping with relatively higher error and unstable performance. These findings highlight *AKRMap*'s superior reliability and robustness across diverse embedding scenarios.

**Ablation results**. The ablation results highlight the impor-

---

[1]https://github.com/mortezamg63/Supervised-autoencoder

tance of the two key components of our method: kernel regression supervision (KR) and the adaptive generalized kernel (GK), both of which are critical for enhancing mapping accuracy. Among these, kernel regression supervision proves to be the most impactful, as the results for the AKRMap *w/o KR* setting are nearly indistinguishable from those of the baseline methods in Table 1. This outcome is expected, as removing kernel regression supervision leaves the projection network relying solely on the neighborhood constraint, making it functionally similar to the t-SNE method. In addition, the adaptive generalized kernel demonstrates its value by further reducing errors across various metrics, underscoring its effectiveness in capturing and fitting complex metric landscapes. These findings validate the necessity of both components in achieving superior performance.

## 5.2. Applications

### 5.2.1. Visualizing Large T2I Dataset

*AKRMap* can provide an accurate overview of large T2I dataset by effectively capturing the cross-modal metric distribution. Figure 5 shows the contour maps of ClipScore, HPSv2, PickScore, and Aesthetic Score distributions in the HPD dataset, generated by our *AKRMap* and four baseline methods: PCA, UMAP, t-SNE, and Neuro-Visualizer.

In alignment with the results from the quantitative experiments, our method demonstrates superior performance by accurately depicting score distributions with rich local details. In contrast, the baselines suffer from various limitations, such as over-smoothing due to averaging effects, which obscure local structures. For example, PCA introduces pronounced block effects with sharp edges at region boundaries, significantly disrupting the overall smoothness of the visualizations. Similarly, UMAP and t-SNE often struggle with low-value regions being either overshadowed by high values or averaged out, making it challenging to identify clusters with suboptimal performance. Surprisingly, Neuro-Visualizer performs the worst among the baselines for visualizing cross-modal embedding metrics. Except the Aesthetic Score, its results are highly unstable, with excessively large estimations in out-of-sample areas, and the contour maps are riddled with jagged terrains, exhibiting poor smoothness. This highlights the increased challenge of cross-modal embeddings compared to unimodal embeddings. It also further underscores the strength of our *AKRMap* in producing more accurate and visually coherent representations. Overall, our *AKRMap* achieves a superior balance between local accuracy and smoothness. Furthermore, the mapping performance of traditional methods can deteriorate significantly if a smaller bandwidth is manually set, as demonstrated by additional results in Appendix C.

### 5.2.2. Comparing diffusion-based model and auto-regressive model

We conduct a visual comparison of two representative T2I models from different architectural families: Stable Diffusion-v2.1 (Rombach et al., 2022) (SD-2.1), a diffusion-based model, and Infinity-2B (Han et al., 2024), an auto-regressive model. Using approximately 590,000 image captions from the MS-COCO (Lin et al., 2014) training dataset as prompts, we generate corresponding images using both models. To illustrate their performance differences, we visualize the HPSv2 score differences between Infinity and SD-2.1 (calculated as Infinity's score minus SD-2.1's score), as shown in Figure 6. Here, red regions highlight areas where Infinity outperforms SD-2.1 significantly, while blue regions indicate smaller differences.

The visualization reveals interesting patterns, particularly in Region A, where a significant performance gap exists between the two models. A deeper analysis reveals that this region is primarily associated with prompts related to sports players and athletes. By examining instance-level generated images with overlay feature, we find that SD-2.1 often struggles to generate human figures or produces black-and-white photographs, whereas Infinity consistently generates high-quality, colored images of athletes; see the right side of Figure 6. This indicates that Infinity demonstrates stronger capabilities in generating human figures, especially in sports-related contexts. Overall, *AKRMap* effectively highlights performance distribution patterns across large-scale datasets, facilitating a detailed comparative analysis of the strengths and limitations of different models.

### 5.2.3. Visualizing Fine-tuned Model

We showcase *AKRMap*'s ability to analyze the impact of model fine-tuning by comparing the SD model (Rombach et al., 2022) with Dreamlike Photoreal 2.0 (DP-2.0) (dreamlike.art, 2024), a fine-tuned variant of SD-1.5 optimized for photorealistic image generation. Our evaluation utilizes two distinct datasets: the MS-COCO (Lin et al., 2014) validation set, from which we extract 25,000 image captions as prompts, and the PartiPrompts (Yu et al., 2022) dataset, which contains 1,600 diverse prompts. Consistent with the previous analysis, we visualize the HPSv2 score differences between DP-2.0 and SD-1.5 to illustrate their comparative performance, with the results presented in Figure 7.

Overall, the effects of fine-tuning are more pronounced on the PartiPrompts benchmark, as evidenced by a larger presence of red regions in the right visualization. This is likely because MS-COCO dataset contains common captions on the web that are similar to the pretraining data of SD model, while PartiPrompts contains manually created sophisticated prompts that are challenging in different aspects. Moreover, the visualizations highlight several regions of interest (A,

CLIPScore

HPSv2

PickScore

Aesthetic Score

PCA UMAP t-SNE Neuro-Visualizer AKRMap (Ours)

*Figure 5.* Qualitative comparison of contour map visualizations of ClipScore, HPSv2, PickScore, and Aesthetic Score distributions in the HPD dataset generated by our *AKRMap* and four baselines: PCA, UMAP, t-SNE, and Neuro-Visualizer.

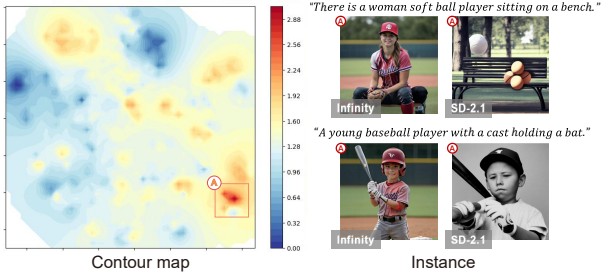

*Figure 6. AKRMap* can be used to compare generative performance of auto-regressive model and diffusion model.

B, and C) where DP-2.0 demonstrates significant improvements in photorealism and overall image quality. In Region A, further analysis of individual instances reveals that DP-2.0 achieves enhanced realism in both static and dynamic objects (such as oranges and flying birds) and shows notable improvements in scenes, such as beach waves and sand textures. Region B primarily includes various stylistic renditions of raccoon images, where DP-2.0 not only accurately

captures the intended styles but also excels in rendering intricate details, such as patterns on ties and textiles. Region C encompasses a variety of automotive scenes, where both the environmental contexts and the vehicles themselves exhibit greater realism and richer detail compared to SD-1.5. Importantly, our visualization demonstrates that DP-2.0 consistently outperforms SD-1.5 across the entire distribution space. This indicates that the fine-tuning process successfully enhanced the model's photorealism while maintaining its general capabilities across other domains.

## 6. Discussion

**Limitations**. As *AKRMap* prioritizes metric mapping, the neighborhood preservation performance may not be comparable to traditional DR methods. Nevertheless, due to the incorporation of neighborhood constraint, *AKRMap* is able to maintain a desirable level of neighborhood preservation, with detailed comparison results shown in Appendix A. Furthermore, for more complex embedding metrics used

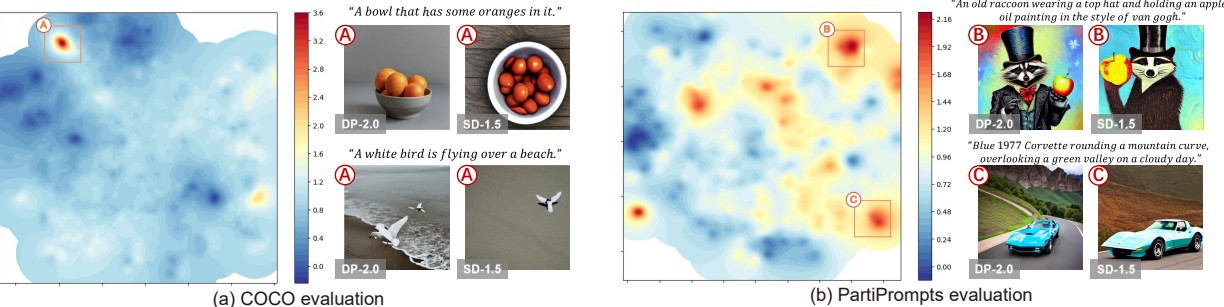

*Figure 7. AKRMap* can be used to show the global impact of fine-tuning by comparison to base model.

in other tasks, such as those computed over sequences of embeddings (e.g., CodeBertScore (Zhou et al., 2023)), it remains uncertain how *AKRMap* can effectively determine a single vector representation for each instance prior to projection. This presents a potential area for further investigation.

**Future Work**. A recent study introduces Multi-dimensional Human Preference (MPS) (Zhang et al., 2024), which evaluates embedding scores across four dimensions: *Overall, Aesthetics, Alignment*, and *Detail*. Leveraging *AKRMap* to visualize multi-dimensional comparisons of different T2I models would be an interesting avenue for future exploration. *AKRMap* offers the potential to enhance existing cross-modal metric models through interactive human feedback. For example, we aim to fine-tune models to address domain-specific needs, such as evaluating and filtering game assets generated by T2I models.

Moreover, our method provides a versatile framework that can be adapted to other value landscape mapping challenges for high-dimensional data. For instance, *AKRMap* could be employed to visualize the distribution of predicted values in classical multivariate regression tasks. Beyond evaluation, *AKRMap* could also play a role in supporting trustworthy filtering of pretraining data by visualizing filtering scores across large-scale datasets. One pertinent example involves the use of Aesthetic Scores to filter the massive LAION-5b dataset (Schuhmann et al., 2022). We plan to scale our approach to datasets containing billions of data points, thereby increasing transparency in pretraining data selection and promoting better understanding of these processes.

## 7. Conclusion

In this paper, we introduce *AKRMap*, a dimensionality reduction method to visualize cross-modal embedding metrics through kernel regression supervised projection with adaptive generalized kernel. Based on *AKRMap*, we develop a visualization tool to support metric-aware visualization of cross-modal embeddings for the evaluation of text-to-image generative models. Quantitative experiments and three application scenarios show that *AKRMap* can facil-

itate trustworthy visualization of cross-modal metric for transparent evaluation.

## Acknowledgments

We would like to extend our gratitude to the anonymous reviewers for their valuable comments. This work is partially supported by the National Natural Science Foundation of China (62172398, U23A20313, 62372471), The Science Foundation for Distinguished Young Scholars of Hunan Province (NO. 2023JJ10080), and Guangzhou Basic and Applied Basic Research Foundation (2024A04J6462).

## Impact Statement

*AKRMap* advances transparent multi-modal evaluation by trustworthy visualization of cross-modal embedding metrics, which is pivotal for enhancing human-in-the-loop evaluation in tasks like T2I generation. The strategy of jointly learning discrete point DR with continuous metric landscape estimation proves highly effective in scenarios like visualizing large generated dataset, comparing different models' performance, and revealing the global impact of fine-tuning. Our method also provides a general framework that can be adapted to embedding-based metric visualization for other tasks such as 3D generation and pretraining data filtering.

Our method highlights the synergy between automatically computed metrics and human's interactive interpretation. With the rapid development of AI models trained and evaluated on datasets of tremendous scale, it is critical to strike a balance between the efficiency and trustworthiness of evaluation, where the transparency and visibility to humans are an essential factor.

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

# Appendix

## A. Performance in Traditional Neighborhood Preservation Objective

Traditional dimensionality reduction methods often seek to preserve the neighborhood in high-dimensional space in the visualization scatterplot. Since the target of our method is not to optimize the neighborhood preservation, we do not expect our method to be better than traditional dimensionality reduction method in this regard. In Table 2, we provide statistics of traditional neighborhood trustworthiness metric (Kaski et al., 2003) to show that our method does not significantly harm the neighborhood property. Furthermore, we note that in our application where we focus on continuous metric, cluster separation is not a meaningful target since we are not looking at discrete categories as in classification tasks. As shown in Figure 8, for the HPSv2 metric on HPD dataset, even in scatterplot view *AKRMap* can reveal the distribution of metric significantly better than other traditional DR methods. Here we also note that we deliberately select larger perplexity for the t-SNE method because of the large scale of the HPD dataset, and we show in Figure 9 that smaller perplexity will lead to worse results.

*Table 2.* Neighborhood Trustworthiness metrics for visualization of HPSv2 embeddings on HPD.

|  | n=20 | n=30 | n=40 | n=50 |
|---|---|---|---|---|
| **PCA** | 0.7243 | 0.7246 | 0.7245 | 0.7242 |
| **t-SNE** | 0.8945 | 0.8786 | 0.8667 | 0.8575 |
| **UMAP** | 0.8604 | 0.8467 | 0.8380 | 0.8309 |
| **AKRMap** | 0.8284 | 0.8287 | 0.8283 | 0.8277 |

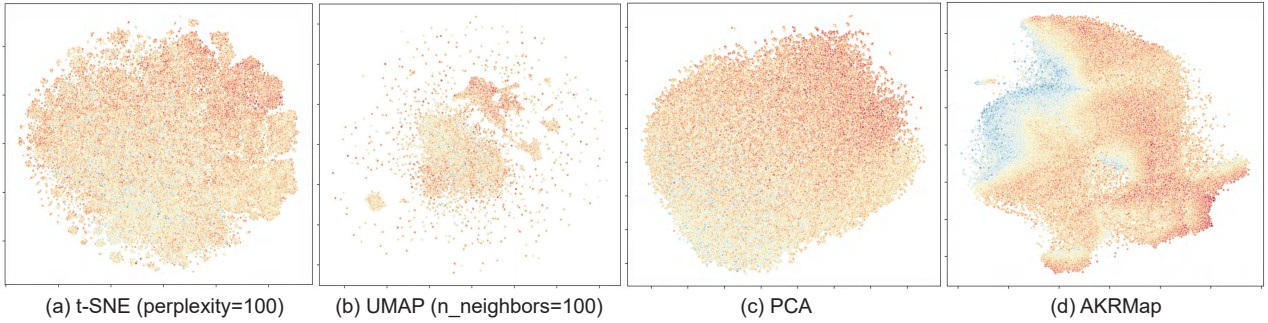

(a) t-SNE (perplexity=100)     (b) UMAP (n_neighbors=100)     (c) PCA     (d) AKRMap

*Figure 8.* Comparison of scatterplots produced by different DR methods for HPSv2.

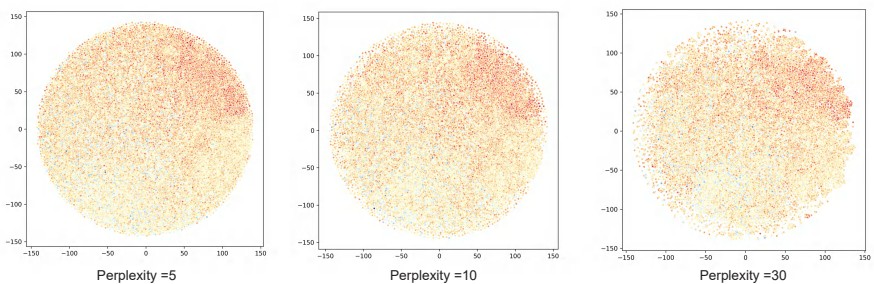

Perplexity =5     Perplexity =10     Perplexity =30

*Figure 9.* Results of smaller perplexity for t-SNE on HPSv2.

## B. Mapping Accuracy Evaluation for in-sample Points

In this section, we record the quantitative results for in-sample mapping accuracy of different methods, as shown in Table 3. The results combined with Table 1 indicate that *AKRMap* outperforms other methods consistently for both in-sample and out-of-sample positions.

*Table 3.* Quantitative comparison of mapping accuracy at in-sample points of HPD dataset.

| Method | CLIPScore | | | HPSv2 | | | PickScore | | | Aesthetic Score | | |
|---|---|---|---|---|---|---|---|---|---|---|---|---|
| | $mae$ | $mape$ | $rmse$ | $mae$ | $mape$ | $rmse$ | $mae$ | $mape$ | $rmse$ | $mae$ | $mape$ | $rmse$ |
| PCA | 4.0494 | 22.8348 | 5.3954 | 1.3744 | 5.3870 | 1.7690 | 1.1341 | 5.7472 | 1.4437 | 0.5209 | 10.6224 | 0.6749 |
| t-SNE | 4.0573 | 22.8162 | 5.3798 | 1.3567 | 5.3259 | 1.7606 | 1.1200 | 5.6774 | 1.4340 | 0.5192 | 10.5890 | 0.6728 |
| UMAP | 4.0906 | 22.8612 | 5.4106 | 1.4755 | 5.7803 | 1.9073 | 1.1804 | 5.9824 | 1.5145 | 0.5241 | 10.6851 | 0.6782 |
| SAE | 4.2591 | 23.2463 | 5.5196 | 1.5323 | 5.9966 | 1.9738 | 1.2731 | 6.4886 | 1.6270 | 0.5207 | 10.6322 | 0.6746 |
| Neuro-Visualizer | 10.4468 | 39.8135 | 12.1020 | 1.8716 | 7.0312 | 2.2237 | 2.0031 | 9.6188 | 2.3844 | 0.3102 | 6.1445 | 0.4009 |
| AKRMap (w/o KR) | 4.0506 | 22.6324 | 5.3415 | 1.3696 | 5.3706 | 1.7761 | 1.1237 | 5.6914 | 1.4343 | 0.3619 | 7.2980 | 0.4654 |
| AKRMap (w/o GK) | 1.6359 | 8.3539 | 2.1271 | 1.0479 | 4.0655 | 1.3358 | 0.9062 | 4.5448 | 1.1537 | 0.3390 | 6.7634 | 0.4320 |
| AKRMap | 1.1652 | 5.4238 | 1.5029 | 0.6834 | 2.6159 | 0.8980 | 0.6815 | 3.4181 | 0.8862 | 0.3142 | 6.1747 | 0.4018 |

## C. RBF Kernel Settings for Traditional DR Methods

Here we provide details and more experiment results concerning RBF kernel settings for traditional projection method. We show in Table 4 and Table 5 that traditional methods cannot achieve satisfactory performance regardless of the kernel parameter settings. Specifically, we test different commonly used bandwidth selection methods:

- **Plug-in method (Silverman)** (Silverman, 2018). This is the default method we used in the paper due to its efficiency. For d-dimensional data, the Silverman's rule is given by:

$$h = \left(\frac{4}{(d+2)n}\right)^{\frac{1}{d+4}} \hat{\sigma}, \tag{7}$$

where $\hat{\sigma}$ is the estimated standard deviation.

- **Adaptive local bandwidth (ALB)** (Cheng & Mueller, 2015). This method computes bandwidth for each point by adapting a pre-computed bandwidth based on local estimated density $f(x_i)$.

$$h_i = \lambda_i \times h, \ \ \lambda_i = (G/f(x_i))^2, \ \ G = \left(\prod_{i=1}^{n} f(x_i)\right)^{1/n}. \tag{8}$$

- **Cross-validation (LOOCV)** (Węglarczyk, 2018). The cross-validation is performed in a Leave-one-out manner:

$$CV(h) = n^{-1} \sum_{j=1}^{n} [y_j - \hat{s}_j(x_j)]^2 w(x_j), \tag{9}$$

where $\hat{s}_j(x_j)$ is the leave-one-out estimator for $y_j$ that is computed on all data points except $x_j$, and $w(x_j)$ is a nonnegative weight function (all set to one by default). Then $h$ is selected to minimize this validation error.

*Table 4.* Quantitative comparison of out-of-sample mapping accuracy of different bandwidth selection methods for traditional DR.

| Method | CLIPScore | | | HPSv2 | | | PickScore | | |
|---|---|---|---|---|---|---|---|---|---|
| | $mae$ | $mape$ | $rmse$ | $mae$ | $mape$ | $rmse$ | $mae$ | $mape$ | $rmse$ |
| PCA+Silverman | 4.0042 | 16.5506 | 5.1142 | 1.4444 | 5.7870 | 1.8749 | 1.1497 | 5.9209 | 1.4686 |
| PCA+ALB | 4.0163 | 16.5445 | 5.1148 | 1.4390 | 5.7662 | 1.8651 | 1.1469 | 5.9140 | 1.4658 |
| PCA+LOOCV | 4.0000 | 16.5140 | 5.1034 | 1.4347 | 5.7492 | 1.8598 | 1.1415 | 5.8769 | 1.4557 |
| t-SNE+Silverman | 4.0241 | 16.6643 | 5.1523 | 1.4361 | 5.7568 | 1.8629 | 1.1579 | 5.9429 | 1.4640 |
| t-SNE+ALB | 4.0215 | 16.6381 | 5.1497 | 1.5147 | 6.1203 | 1.9546 | 1.1463 | 5.8838 | 1.4524 |
| t-SNE+LOOCV | 4.0205 | 16.6276 | 5.1366 | 1.5250 | 6.1516 | 1.9573 | 1.1520 | 5.9210 | 1.4639 |
| UMAP+Silverman | 4.0819 | 16.9066 | 5.2179 | 1.4432 | 5.7817 | 1.8725 | 1.2214 | 6.2721 | 1.5536 |
| UMAP+ALB | 4.0142 | 16.7284 | 5.1332 | 1.4171 | 5.6262 | 1.8285 | 1.2019 | 6.1807 | 1.5276 |
| UMAP+LOOCV | 4.0417 | 16.7337 | 5.1617 | 1.4830 | 5.9180 | 1.9233 | 1.2715 | 6.5419 | 1.6274 |

In addition, we also show that it is not possible to achieve accurate mapping by manually setting smaller bandwidth than the automatically selected one. In fact, this will evidently cause severe overfitting issue and result in fragmented blocks in the map, as shown in Figure 10 of t-SNE method. This also causes the quantitative accuracy of the mapping to drop. For example, for the case in Figure 10, the error has increased as shown in Table 6.

*Table 5.* Quantitative comparison of in-sample mapping accuracy of different bandwidth selection methods for traditional DR.

| Method | CLIPScore | | | HPSv2 | | | PickScore | | |
|---|---|---|---|---|---|---|---|---|---|
| | *mae* | *mape* | *rmse* | *mae* | *mape* | *rmse* | *mae* | *mape* | *rmse* |
| PCA+Silverman | 4.0494 | 22.8348 | 5.3954 | 1.3744 | 5.3870 | 1.7690 | 1.1341 | 5.7472 | 1.4437 |
| PCA+ALB | 4.0415 | 22.8072 | 5.3806 | 1.3702 | 5.3707 | 1.7621 | 1.1284 | 5.7254 | 1.4394 |
| PCA+LOOCV | 4.0505 | 22.9057 | 5.3927 | 1.3747 | 5.3874 | 1.7674 | 1.1351 | 5.7502 | 1.4447 |
| t-SNE+Silverman | 4.0573 | 22.8162 | 5.3798 | 1.3567 | 5.3259 | 1.7606 | 1.1200 | 5.6774 | 1.4340 |
| t-SNE+ALB | 4.0476 | 22.7261 | 5.2578 | 1.3549 | 5.3168 | 1.7564 | 1.1164 | 5.6626 | 1.4292 |
| t-SNE+LOOCV | 4.0447 | 22.6818 | 5.3511 | 1.3545 | 5.3158 | 1.7569 | 1.1131 | 5.6390 | 1.4270 |
| UMAP+Silverman | 4.0906 | 22.8612 | 5.4106 | 1.4755 | 5.7803 | 1.9073 | 1.1804 | 5.9824 | 1.5145 |
| UMAP+ALB | 4.0675 | 22.9090 | 5.3831 | 1.4621 | 5.7202 | 1.8871 | 1.1729 | 5.9458 | 1.5051 |
| UMAP+LOOCV | 4.0684 | 22.7450 | 5.3680 | 1.4699 | 5.7576 | 1.9013 | 1.1730 | 5.9430 | 1.5083 |

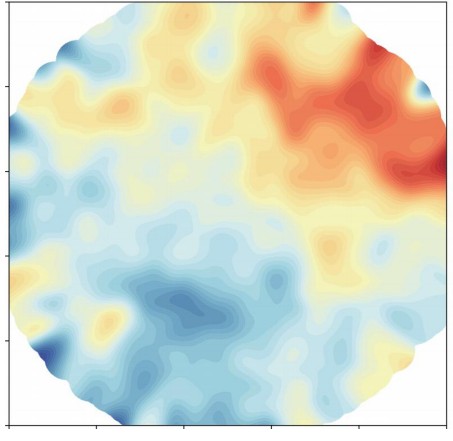
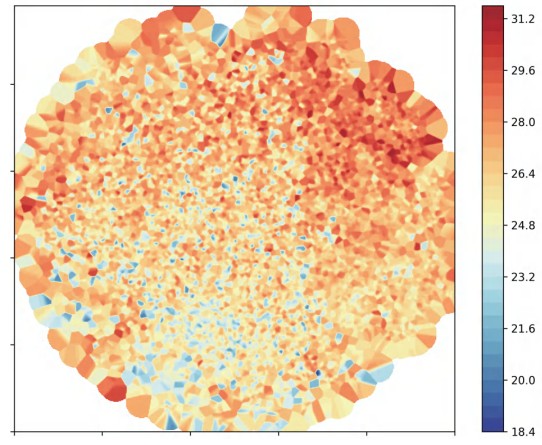

(a) auto-selected bandwidth          (b) manually set 0.1x bandwidth

*Figure 10.* Attempts to manually selecting smaller bandwidth for traditional methods may lead to worse mapping performance with severe overfitting, as exemplified by the t-SNE visualization of HPSv2 score distribution.

## D. Zoom in by Different Grid Numbers

This section shows the zoom-in effect of *AKRMap* when increasing the grid number. As shown in Figure 11, *AKRMap* can accurately estimate the local distribution of cross-modal metric and adds detail on demand. Specifically, with a small grid number of 30 by 30, *AKRMap* can already achieve the level of detail close to most baseline methods with grid number of 500 by 500. When increasing the grid number, we can see in Figure 11 that our method can accurately show the detail contour of metric distribution.

## E. Hyperparameter

In this section we discuss some hyperparameter trade-off in our method with additional experimental results. First, we show the trade-off effect by different $\lambda$ value to weight the regression loss and KL loss, with test (out-of-sample) quantitative results shown in Table 7, and visualization effects shown in Figure 12. Next, we show the scatterplot view of our ablation studies in Figure 13 to demonstrate the effect of our proposed component in large-scale DR point plot. Finally, in Figure 14, we illustrate some qualitative and quantitative effect of setting the train-val weights $w_1$ or $w_2$ to be zero to demonstrate the necessity of the train-val balance scheme in our kernel regression guidance loss.

*Table 6.* Impact of manually setting smaller bandwidth for Figure 10.

| | $mae_{in}$ | $mape_{in}$ | $rmse_{in}$ | $mae_{out}$ | $mape_{out}$ | $rmse_{out}$ |
|---|---|---|---|---|---|---|
| **auto-selected bandwidth** | 1.3567 | 5.3259 | 1.7606 | 1.4361 | 5.7568 | 1.8629 |
| **0.1x bandwidth** | 1.5403 | 6.0284 | 2.0470 | 1.7050 | 6.7994 | 2.1951 |

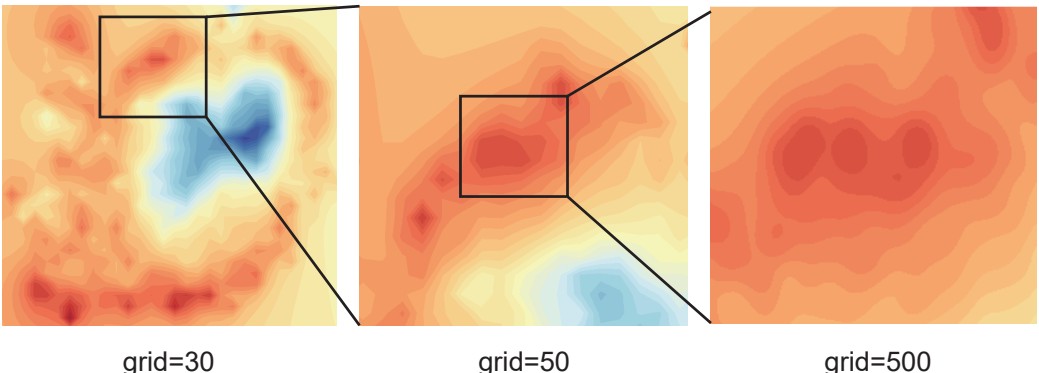

grid=30        grid=50        grid=500

*Figure 11.* Zoom-in effect by different grid numbers in our method.

*Table 7.* Experimental results on trade-off effect by $\lambda$ settings.

| Method | CLIPScore | | | HPSv2 | | | PickScore | | | Aesthetic Score | | |
|---|---|---|---|---|---|---|---|---|---|---|---|---|
| | $mae$ | $rmse$ | $trust$ | $mae$ | $rmse$ | $trust$ | $mae$ | $rmse$ | $trust$ | $mae$ | $rmse$ | $trust$ |
| $\lambda = 0.05$ | 2.0752 | 2.6639 | 0.7849 | 1.3613 | 1.7624 | 0.8635 | 1.1048 | 1.4027 | 0.8457 | 0.4625 | 0.5869 | 0.8731 |
| $\lambda = 0.125$ | 1.8707 | 2.4253 | 0.7672 | 0.8108 | 1.1200 | 0.8284 | 0.7712 | 1.0225 | 0.8143 | 0.4305 | 0.5488 | 0.8704 |
| $\lambda = 0.25$ | 3.4395 | 4.4237 | 0.6865 | 0.5621 | 0.8048 | 0.8111 | 0.5511 | 0.7413 | 0.7990 | 0.4039 | 0.5149 | 0.8626 |
| $\lambda = 0.5$ | 3.4811 | 4.4620 | 0.7083 | 0.4453 | 0.6554 | 0.7847 | 0.4597 | 0.6346 | 0.7791 | 0.3875 | 0.4925 | 0.8545 |

## F. Test on Other Modality Embeddings

We further perform quantitative experiments on CLIP text-video embeddings on MSR-VTT dataset (Xu et al., 2016) which consists of 200,000 text-video pairs (Table 10), as well as text-audio embeddings and image-audio embeddings on Flickr 8k Audio Caption Corpus dataset (Harwath & Glass, 2015) consisting of 40,000 pairs (Table 9 and Table 8). Overall, the results indicate that our method can improve upon traditional DR in test set mapping accuracy.

*Table 8.* ImageBind image-audio embeddings mapping accuracy on Flickr 8k Audio Caption Corpus.

| | PCA | t-SNE | UMAP | AKRMap |
|---|---|---|---|---|
| $mae$ | 4.1239 | 3.9576 | 3.8811 | 1.9425 |
| $mape$ | 19.1321 | 18.3808 | 18.1024 | 8.9521 |
| $rmse$ | 5.1737 | 5.0254 | 4.8633 | 2.6425 |

## G. Weight Balancing Mechanism

We also test automatic weight balancing method, using a sigmoid function to adjust the weight of KL loss.

$$w(x, \mu) = \sigma(k(x - \mu)) = \frac{1}{1 + e^{-k(x-\mu)}}, \tag{10}$$

where $\mu$ is a threshold of acceptable $KL$ loss which we set to 2 by default and $k$ is a parameter to control the decay rate which we default to 1. Then the total loss becomes:

$$L_1 = \lambda MSE_r + w(KL, \mu) \cdot KL, \tag{11}$$

where $\lambda$ is fixed to the default value 0.125.

Alternatively, for users who care more about the neighborhood preservation, we can use this method to weight the MSE term with a user-specified threshold $\mu_1$:

$$L_2 = w(MSE_r, \mu_1) \cdot \lambda MSE_r + \cdot KL. \tag{12}$$

The experimental results for different $\mu$ and $\mu_1$ settings are presented in Table 11.

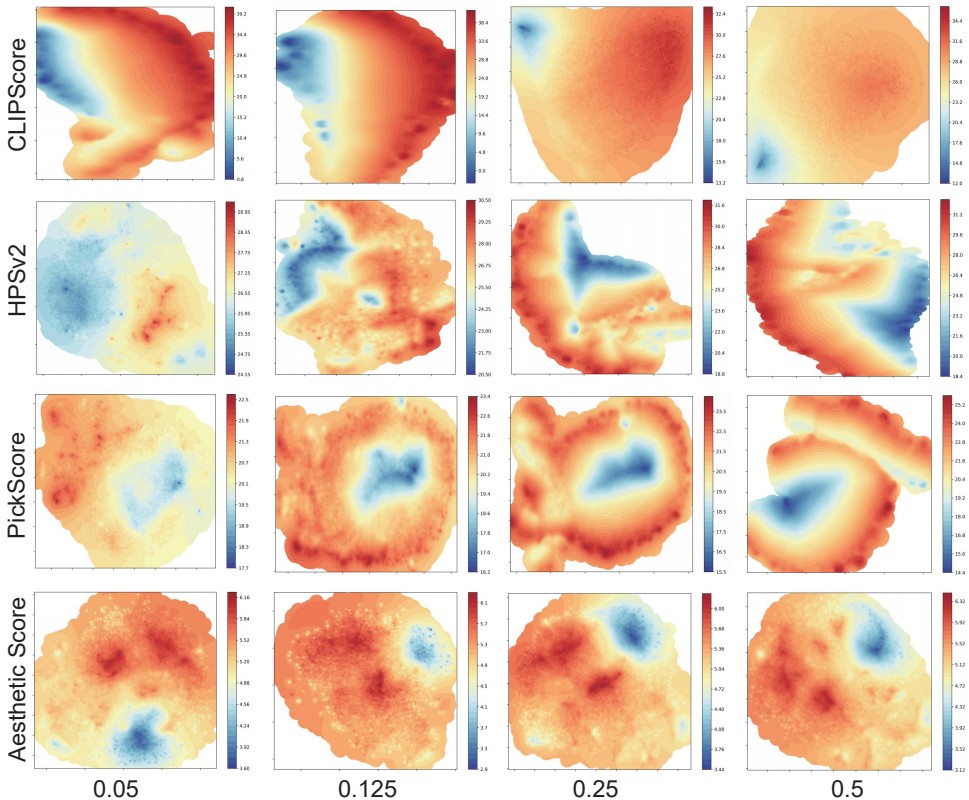

*Figure 12.* AKRMap mapping results for different $\lambda$ settings.

*Table 9.* ImageBind text-audio embeddings mapping accuracy on Flickr 8k Audio Caption Corpus.

|  | PCA | t-SNE | UMAP | AKRMap |
|---|---|---|---|---|
| *mae* | 3.7194 | 3.5992 | 3.6801 | 1.8629 |
| *mape* | 21.1113 | 20.3858 | 20.8498 | 10.1821 |
| *rmse* | 4.7175 | 4.5607 | 4.6541 | 2.4841 |

## H. Interactive Features Evaluation

We conduct a user study among 12 users on our interactive features including zoom and overlay. They complete a 7 Likert-scale survey after using our interactive visualization of HPSv2 score on HPD dataset. As shown in Figure 15, users generally appreciate our interactive features in several usability aspects including effectiveness, ease of use, interpretability, and future use.

## I. Convergence Properties of the Adaptive Generalized Kernel Regression

In this appendix, we summarize the theoretical conditions and practical considerations related to the convergence of our adaptive generalized kernel regression method, which extends the classical Nadaraya-Watson kernel regression framework.

To guarantee convergence in terms of mean squared error (MSE), the following conditions are essential:

- **1) Integrability of the generalized kernel on $\mathbb{R}^2$:** The kernel is defined as:

$$K(x) = (1 + \alpha|x|^{2\beta})^{-1}. \tag{13}$$

  For $K(x)$ to be integrable and finite over $\mathbb{R}^2$, parameters must satisfy $\alpha > 0$ and $\beta > 1$, ensuring sufficient decay of the kernel function at infinity.

- **2) Growth behavior of the parameter $\alpha$ with sample size $N$:** The parameter $\alpha$ must effectively increase as the

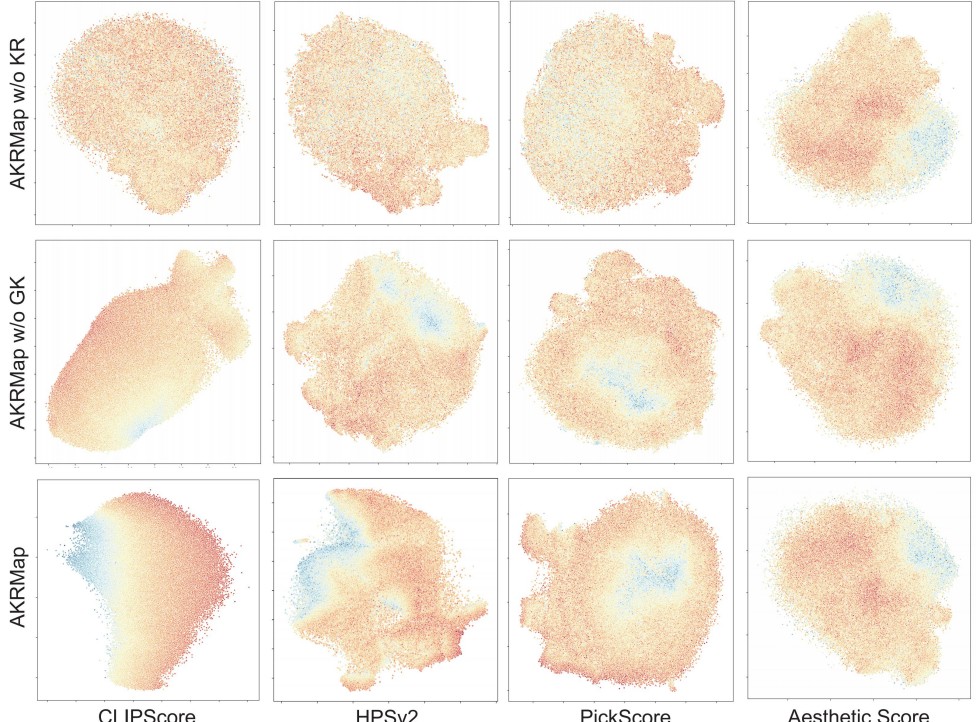

*Figure 13.* Scatterplots of ablation.

*Table 10.* CLIP text-video embeddings mapping accuracy on MSR-VTT datasets.

|  | **PCA** | **t-SNE** | **UMAP** | **AKRMap** |
|---|---|---|---|---|
| $mae$ | 1.4908 | 1.4201 | 1.4562 | 0.6763 |
| $mape$ | 8.3080 | 7.9589 | 8.1636 | 3.4832 |
| $rmse$ | 2.1231 | 2.0216 | 2.0723 | 0.9578 |

sample size $N$ grows, analogous to the classical kernel bandwidth $h$ decreasing with larger $N$. This condition is critical for establishing consistency and uniform convergence of the estimator.

- **3) Smoothness of the regression target function** $s(x)$**:** The target function $s(x)$ should satisfy smoothness assumptions such as Lipschitz continuity. This is a standard requirement in kernel regression theory to control the bias term and facilitate uniform convergence as bandwidth parameters are tuned.

In our training procedure, we enforce the non-negativity of parameters $\alpha$ and $\beta$ through reparameterization (by squaring them) and include them within a set of learnable parameters updated via backpropagation. Intuitively, this process pushes the network to automatically identify suitable decay rates for the kernel in each epoch or mini-batch, adapting progressively to match the magnitude scale of the Nadaraya-Watson kernel. Empirical experiments demonstrate that the learned parameter values $(\alpha, \beta)$ converge successfully. For example, the values for HPSv2 are (68.57, 1.61), for PickScore are (59.13, 1.35), for CLIPScore are (104.70, 3.18), and for Aesthetic Score are (74.95, 1.11). These results indicate that our data-driven approach effectively satisfies conditions (1) and (2) mentioned above.

In practical multimodal scenarios, although the target function might not strictly satisfy Lipschitz continuity, numerical experiments indicate that kernel regression methods remain robust as long as the target distribution does not exhibit extreme irregularities, such as highly discontinuous or severely jagged patterns. In other words, even piecewise smooth or locally continuous target functions can yield stable and accurate kernel regression estimates in practice. Specifically, in our scenario involving high-dimensional to low-dimensional mappings, there exists an continuous relationship between evaluation metrics and the projected cross-modal embeddings. For example, CLIPScore, which is computed using cosine similarity between image and text embeddings, is an obvious continuous function between the original high-dimensional embeddings and the metric. Furthermore, our parametric projection network is an explicit continuous and differentiable function. The Implicit

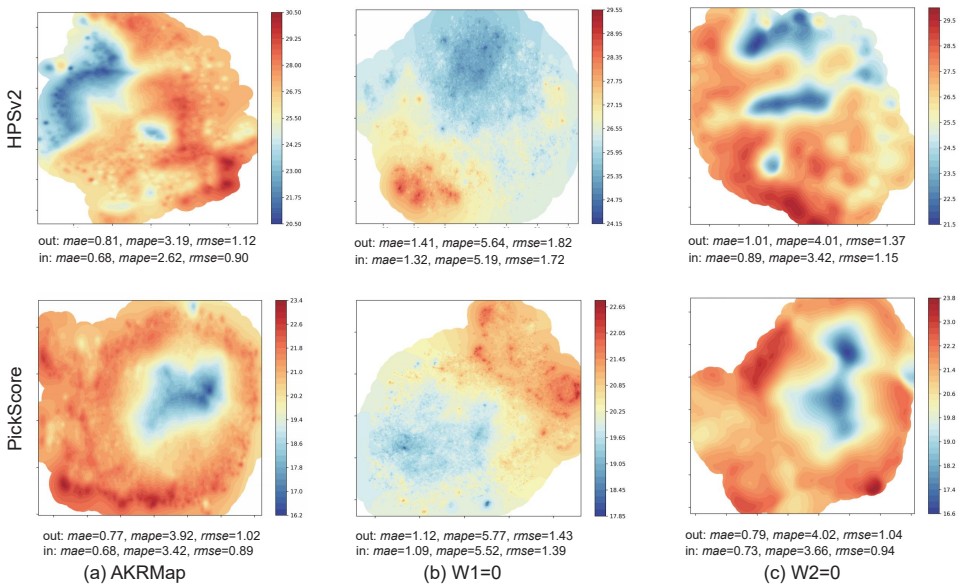

*Figure 14.* Comparison with setting $w_1 = 0$ and $w_2 = 0$ in our MSE loss on HPSv2.

*Table 11.* Weight balancing results of $L_1$ and $L_2$ on HPSv2.

|  | $mae$ | $mape$ | $rmse$ | $trust$ |
|---|---|---|---|---|
| w/o weight balance | 0.8108 | 3.1935 | 1.1200 | 0.8284 |
| $\mu = 2$ | 0.5118 | 2.0318 | 0.7464 | 0.8053 |
| $\mu = 1.8$ | 0.4963 | 1.9632 | 0.7051 | 0.7979 |
| $\mu = 1.6$ | 0.5336 | 2.1115 | 0.7772 | 0.8059 |
| $\mu_1 = 1$ | 1.0427 | 4.1393 | 1.3787 | 0.8399 |
| $\mu_1 = 0.8$ | 0.9610 | 3.8069 | 1.2686 | 0.8316 |
| $\mu_1 = 0.6$ | 1.0118 | 1.9999 | 1.3478 | 0.8426 |

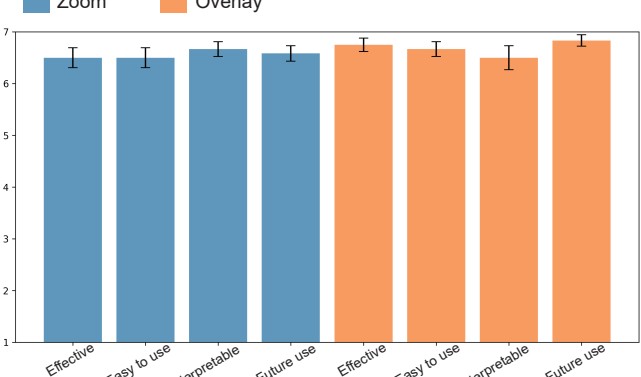

*Figure 15.* User study results on interactive features.

Function Theorem can then ensure the continuity of the metric $w.r.t$ the projected 2D embeddings. Thus, our practical setup implicitly fulfills the smoothness assumption (condition 3) required for convergence.

Furthermore, the mapping function $s(x)$ is continuously updated during dimensionality reduction (projection). Our adaptively updated generalized kernel not only naturally accommodates complex distributions encountered in practice but also ensures consistency in metric spaces through supervised kernel regression. Regarding projection stability, our kernel regression loss can be regarded as a regularization term that enforces the global structure of the projection, as it explicitly pushes the projected sample points to regress a global metric distribution.

