# OpenReview forum: "AKRMap: Adaptive Kernel Regression for Trustworthy Visualization of Cross-Modal Embeddings"
_ICML.cc/2025/Conference — ICML 2025 poster_

### Official Review · Reviewer_63Bw · 2025-02-21

**Overall Recommendation:** 3

**Summary:**

In their paper "AKRMap: adaptive kernel regression for trustworthy visualization of cross-modal embeddings", the authors suggest what I would call *supervised parametric t-SNE* where supervision is by a continuous variable (as opposed to discrete classes). The algorithm combines the t-SNE loss function with a kernel regression loss function, to ensure that the resulting 2D embedding allows to predict the continuous variable of interest. The authors then apply this to visualize concatenated high-dimensional embeddings of (text prompt, generated image) pairs, colored by the prompt-image alignment metric (which provides the supervised signal). They authors argue that the resulting visualizations are helpful to get insight into the performance of image-generating models.

**Claims And Evidence:**

Yes.

**Essential References Not Discussed:**

No.

**Experimental Designs Or Analyses:**

Yes.

**Methods And Evaluation Criteria:**

Yes.

**Other Comments Or Suggestions:**

MINOR COMMENTS

* Section 2: parametric t-SNE should cite https://openreview.net/forum?id=B8a1FcY0vi

* Table 1: everything could be easily rounded to 1 decimal digit

**Other Strengths And Weaknesses:**

Strengths: The algorithm is novel and seems to perform well. The applications are interesting and do provide curious insights.

Weaknesses: The algorithm is quite simple but appears under-studied. In particular, I am missing more systematic hyper-parameter studies and comparisons. All contour maps would be much less confusing if they excluded empty regions.

Methodologically this is a borderline paper (and based only on methodology I would rather tend to rejection), but I do like the applications so am tending towards to acceptance.


MAJOR COMMENTS

* Why do contour maps not exclude empty embdding regions? E.g. in Figure 1, the panel (a) contains some white space, esp. in the corners but also in between t-SNE islands. I would expect the contour map in panel (c) to contain white space in the same places. Instead, the contour map is the full square. This is extremely confusing. Your contour maps are based on a square grid, so can't you make almost-empty squares white? This applies to all contour maps shown in the paper.

* Section 4.1.1: why do you combine training and validation losses? This sounds like a weird approach. When training a model, the loss is usually computed on the training set. Why do you need the validation set here at all? What would you get if you set w_1 = 0 or alternatively w_2 = 0?

* What is the t-SNE perplexity that you use in the KL loss in Eq 5? This is never specificed. Furthermore, you use batch size 1000, but your dataset size is 430,000. The t-SNE's KL loss includes a global normalization, so it's unclear how you can optimize it with mini-batches. Do you compute the p_ij and q_ij t-SNE's affinities (which enter the KL loss) based on each batch separately? If so, and if you use e.g. perplexity 30, then your lambda=0 setting should be similar to t-SNE with effective perplexity 30 * 430,000/1000 = 12,900. Is that right? This needs to be discussed. Note that there is a way to optimize proper t-SNE (with p_ij computed globally for the entire dataset) with minibatches, see e.g. https://openreview.net/forum?id=B8a1FcY0vi, but I assume that's not what you do.

* It would be nice to see AKRMap visualizations for different values of lambda, interpolating from KL to KR, ideally with some metrics like MSE and trustworthiness computed for each lambda and showing a trade-off.

* Ablations: it would be nice to see visualizations like in Figure 3 / Figure 8 for AKRMap w/0 KR and w/o GR. Also, what does "without GR" mean exactly? That you set alpha=beta=1 instead of optimizing these two parameters? By the way, what did alpha and beta converge to in your experiments?

* In Figure 6 and 7 which show score differences, it would be good to use a diverging colorscale centered at 0. Figure 6 should have a colorbar.

* Figure 7a: two example promts seem very different and have nothing in common. Why do they appear so close in the embedding?

**Questions For Authors:**

See above

**Relation To Broader Scientific Literature:**

The paper is related to recent literature on multimodal embeddings and text/image similarity scores.

**Theoretical Claims:**

There are none.

---

> ### Author Rebuttal · Authors · 2025-03-31
>
> Thank you very much for the great review helping us improve the work. We respond to your questions below and provide the figures you ask for in the link: https://docs.google.com/document/d/e/2PACX-1vROjYSQAj8XMbrghXX4Ba_JrAxdFNagPOB84DtTK2PN5B_Ir-CAWew_c6zfhod7h3bU0VgCj_K8zuz8/pub
>
> ## Q1 Why do contour maps not exclude empty embedding regions?
>
> A1: We have fixed the contour maps with a distance threshold to leave white space as you suggested and you can see the effect such as from Fig. 1 to Fig.5 in the link. We will make sure we fix all the figures in the updated version.
>
> ## Q2 Why do you combine training and validation losses?
>
> A2: Thank you. We want to explain that unlike pure neural network methods, kernel regression is nonparametric, where the parameter like bandwidth is traditionally either precomputed or optimized via cross-validation. Thus, we are facing an interesting problem of how to bridge between parametric projection and nonparametric kernel regression, motivating us to set a validation set to learn kernel parameters jointly with the neural network. The validation loss improves the generalization of kernel regression for unseen positions. $w_1$=0 will shrink the kernel to training points with a sharp map and small peaks, while $w_2$=0 will make the map smoother but reduce local detail. Figure 4 in our link displays these two extreme settings and shows that quantitative results may deteriorate when either $w_1$ or $w_2$ is set to zero.
>
> ## Q3 What is the t-SNE perplexity that you use in the KL loss in Eq 5?
>
> A3: Thank you for pointing out the confusion. We actually use a perplexity-free method proposed in this paper: Perplexity-free t-SNE and twice Student tt-SNE, which can compute $\sigma$ using multi-scale Gaussian similarity without manual perplexity. This method can also compute the affinities on batches. We will clarify this in the updated version with reference to recent methods including your suggested paper.
>
> ## Q4 It would be nice to see AKRMap visualizations for different values of lambda.
>
> A4: Thank you for the suggestion. We have added experiments varying lambda between 0.05 and 0.5 to show the trade-off, with both quantitative results and visualizations. The visualization is shown in Fig. 5 in our link. The quantitative results are presented as follows, showing mae against neighbor trustworthiness (n=20). We can see clear trade-off effect ($\lambda$ reduces mae but also decreases neighborhood trust) except in CLIPScore where both low and high $\lambda$ are problematic.
> ### Table 1. trade-off
> |Method| Metric|λ=0.05|λ=0.125|λ=0.25|λ=0.5|
> |-|-|-|-|-|-|
> |**HPSv2**|MAE|1.36|0.81|0.56|0.45|
> || Trust  |0.86|0.83|0.81|0.78|
> |**PickScore**| MAE    | 1.10   | 0.77   | 0.55   | 0.46   |
> ||Trust|0.85|0.81|0.80|0.78|
> |**Aesthetic**|MAE|0.46|0.43|0.40|0.39|
> ||Trust|0.87|0.87|0.86|0.85|
> |**CLIPScore**|MAE|2.08|1.87|3.44|3.48|
> ||Trust|0.78|0.77|0.69|0.71|
>
> ## Q5 Ablations: it would be nice to see visualizations like in Figure 3 / Figure 8 for AKRMap w/0 KR and w/o GR. Also, what does "without GR" mean exactly? What did alpha and beta converge to in your experiments?
>
> A5: Yes, w/o GK means fixing those parameters to 1. We have provided a figure comparing the scatterplots of the ablation results in our link (Fig. 6). In addition, the learned (alpha, beta) values are - HPSv2: (68.57, 1.61), PickScore: (59.13, 1.35), CLIPScore: (104.70, 3.18), Aesthetic Score: (74.95, 1.11).
>
> ## Q6 In Figure 6 and 7 which show score differences, it would be good to use a diverging colorscale centered at 0. Figure 6 should have a colorbar.
>
> A6: Thank you. We have added the colorbar in a revised Figure 6 here in our link (Fig. 2). However, we originally only intended to use color to indicate high and low values instead of sign and we want to explain why we did not center colorscale at 0: the differences between models can be uneven on positive and negative axes. For example, the original Fig. 6 shows an extreme case comparing a 2022 SD model with a quite recent VAR model, where the recent model almost always beats the old one (with sparse outliers). We provide here another two examples of more comparable models: 1) SD3 and SDXL (Fig. 7 in our link), and 2) a new image-to-3D model comparison (Fig.8 in our link). In these examples we can see both positive and negative regions.
>
> ## Q7 Figure 7a: two example promts seem very different and have nothing in common. Why do they appear so close in the embedding?
>
> A7: Thank you. This is possibly because the embedding we project is a concatenation of text and image embeddings. Even though the prompts are different, the images do seem to have similar background features.
>
> ## Q8 Section 2: parametric t-SNE should cite https://openreview.net/forum?id=B8a1FcY0vi
> A8: Thank you. We will add the reference in the updated version.
>
> ## Q9 Table 1: everything could be easily rounded to 1 decimal digit
> A9: Thank you. We will reduce the digit in the updated version.

---

> > ### Comment · Reviewer_63Bw · 2025-04-02
> >
> > Thank you for your responses. I appreciated all the additional figures. I keep my score at 3, and would recommend (weak) acceptance. As all scores are >= 3, it looks promising.
> >
> > Two parts where I did not fully understand your response:
> >
> > > We want to explain that unlike pure neural network methods, kernel regression is nonparametric, where the parameter like bandwidth is traditionally either precomputed or optimized via cross-validation. Thus, we are facing an interesting problem of how to bridge between parametric projection and nonparametric kernel regression, motivating us to set a validation set to learn kernel parameters jointly with the neural network. The validation loss improves the generalization of kernel regression for unseen positions.
> >
> > Just to clarify: does this mean that in the situation where kernel parameters alpha and beta are held fixed (e.g. at alpha=beta=1 values) you would not need the validation loss and could set w1 to zero? If yes, you should clarify this in the paper.
> >
> > > the learned (alpha, beta) values are - HPSv2: (68.57, 1.61), PickScore: (59.13, 1.35), CLIPScore: (104.70, 3.18), Aesthetic Score: (74.95, 1.11).
> >
> > I notice that beta is not very far from 1, but the alpha is always very large. This is likely because your t-SNE loss yields a certain scale of the embedding, and then the kernel works the best if has a matching bandwidth. I am wondering if you could get an equally good performance holding beta=1 and only learning alpha, or even holding both fixed but setting beta=1 and alpha=50 (or some similar value).
> >
> > > We actually use a perplexity-free method proposed in this paper: Perplexity-free t-SNE and twice Student tt-SNE, which can compute using multi-scale Gaussian similarity without manual perplexity. This method can also compute the affinities on batches.
> >
> > These perplexity-free approaches usually use a wide range of perplexities going all the way up to the sample size of the data, and then average the p_ij affinities. I don't understand how this works with batches of size 1000.
> >
> > Also, your Figure 3 shows a comparison to t-SNE with perplexity 100, but what you also should show is a comparison to "Perplexity-free t-SNE" (if that's what you use inside AKRMap).

---

> > > ### Author Response · Authors · 2025-04-06
> > >
> > > Thank you very much for the timely acknowledgement and additional comments. Sorry for a bit of delay in my response as I was waiting for other comments and carefully preparing answers to your additional questions. I hope our answers can address some of your questions.
> > >
> > > ## Q1: Just to clarify: does this mean that in the situation where kernel parameters alpha and beta are held fixed (e.g. at alpha=beta=1 values) you would not need the validation loss and could set w1 to zero? If yes, you should clarify this in the paper.
> > > A1: Yes. We originally set this train-val scheme mainly for more robust learning of kernel parameters in the projection process. If there is no kernel parameter to learn, this scheme is less useful. However, when we further reflect on the problem, we find the situation may be a little more complex here since the regression supervision also affects the projection, which will also in turn impact the post-projection mapping accuracy even in the absence of kernel parameters. We show an additional experiment where we fix $\alpha=\beta=1$ (equivalent to AKRMap w/o GK) and compare the results of keeping w1 and setting w1=0. We find that even under this condition, the validation loss can still make a difference as it can push the projection to adapt to the fixed kernel in a way that avoids overfitting in mapping. Therefore, our preliminary findings is that as long as the kernel regression supervision still exists, the val loss might play a role for more robust mapping. We will add clarification to explain all this in the paper.
> > >
> > > ### Table 1. test accuracy for HPSv2 mapping.
> > > ||$\alpha=\beta=1$|$\alpha=\beta=1, w_1=0$|
> > > |-|-|-|
> > > |mae|1.14|1.39|
> > > |mape|4.55|5.61|
> > > |rmse|1.47|1.81|
> > >
> > > ### Table 2. test accuracy for PickScore mapping.
> > > ||$\alpha=\beta=1$|$\alpha=\beta=1, w_1=0$|
> > > |-|-|-|
> > > |mae|0.91|1.11|
> > > |mape|4.62|5.72|
> > > |rmse|1.17|1.41|
> > >
> > > ## Q2: I am wondering if you could get an equally good performance holding beta=1 and only learning alpha, or even holding both fixed but setting beta=1 and alpha=50 (or some similar value).
> > > A2: Thank you. We have performed additional experiments to test your hypothesis on the HPSv2 embeddings. It seems the slight change in the exponential parameter can still make a difference, possibly because of the sensitivity to exponential change. Directly setting $\alpha$ to a number (50) close to our learned parameter can work better, but still not as good as the original one with fully tunable parameters. In addition, it seems a bit unfair to initialize the parameter close to a learned result since we originally did not have a good reason to guess at a proper scale that can balance between mapping accuracy and neighborhood.
> > > ### Table 3. test accuracy for HPSv2 mapping.
> > > ||$\beta=1$|$\beta=1, \alpha=50$|original|
> > > |-|-|-|-|
> > > |mae|1.06|1.02|0.81|
> > > |mape|4.23|4.04|3.19|
> > > |rmse|1.39|1.34|1.12|
> > >
> > >
> > >
> > >
> > > ## Q3: These perplexity-free approaches usually use a wide range of perplexities going all the way up to the sample size of the data, and then average the p_ij affinities. I don't understand how this works with batches of size 1000.
> > > A3: Thank you very much for this question, which is insightful and further stimulates us to carefully reflect on our implementation of parametric t-SNE. We have truly benefited a lot from your comments. Our implementation of the perplexity-free KL loss follows a github repo: https://github.com/Academich/parametric_tsne_pytorch which uses the multi-scale Gaussian similarities method in my previously suggested paper on each batch. It is similar to your given example, using a wide range of target entropy parameters to compute multiple Gaussian similarities before averaging to get the HD similarities. However, your observation is perfectly correct that when this is only performed in batches, it does not fully accurately capture the affinities on the whole dataset. In this way, our implementation learns to keep the neighboring relationship in each randomly sampled batch and seeks to approximate the global affinities through stochastic gradient descent. In effect, this way tends to emphasize the global structure while indeed preserving the exact neighborhood less accurately. Finally, we are still digesting your suggested paper’s mini-batch method and trying to integrate it into our method for better affinity estimation and better trade-off with our mapping accuracy.
> > >
> > > ## Q4: Also, your Figure 3 shows a comparison to t-SNE with perplexity 100, but what you also should show is a comparison to "Perplexity-free t-SNE" (if that's what you use inside AKRMap).
> > > A4: Thank you for the suggestion. We will add to Figure 3 the scatterplot of our implementation of perplexity-free parametric t-SNE, which is equivalent to the result of AKRMap w/o KR ablation as you can see in the first row of Fig. 5 in my previously provided link.

---

### Official Review · Reviewer_eXC7 · 2025-03-14

**Overall Recommendation:** 3

**Summary:**

The paper proposes AKRMap, an adaptive kernel regression-based dimensionality reduction (DR) method for trustworthy visualization of cross-modal embeddings. Unlike conventional DR techniques (e.g., PCA, t-SNE, UMAP), AKRMap jointly learns a supervised projection network and an adaptive kernel regression loss, improving visualization accuracy of cross-modal metric landscapes. It integrates a generalized adaptive kernel, enabling enhanced contour mapping of evaluation metrics like CLIPScore and Human Preference Score (HPSv2). Empirical results demonstrate AKRMap's superior visualization fidelity over existing methods, facilitating human-in-the-loop analysis of text-to-image (T2I) models, comparative model assessment, and fine-tuning impact evaluation.

**Claims And Evidence:**

The paper's claims are mostly well-supported by quantitative results and qualitative case studies. AKRMap's superior visualization accuracy is validated through lower mapping errors (MAE, RMSE, MAPE) across multiple metrics. The ablation study confirms the necessity of kernel regression supervision and adaptive generalized kernels. However, the claim that AKRMap generalizes to all cross-modal metric landscapes lacks broad empirical support beyond text-to-image embeddings. Further validation on diverse tasks (e.g., speech-text, video-text embeddings) would strengthen this claim. Additionally, while AKRMap improves metric contour accuracy, its neighborhood preservation is weaker than traditional DR methods, which may impact interpretability.

**Essential References Not Discussed:**

The paper lacks discussion on kernel-based DR methods like Kernel PCA (Schölkopf et al., 1998) and Density-Preserving Visualization (Narayan et al., 2021), which relate to its adaptive kernel regression approach. Additionally, multi-modal embedding alignment methods (e.g., ALIGN (Jia et al., 2021)) could provide more context for evaluating cross-modal metrics. While it cites supervised t-SNE (Gisbrecht et al., 2015), newer metric-aware DR techniques (e.g., Triplet UMAP (Sainburg et al., 2021)) are missing. Including these references would strengthen the theoretical foundation and contextualize AKRMap’s contributions.

**Experimental Designs Or Analyses:**

The experimental design is generally sound, with quantitative evaluations (MAE, RMSE, MAPE) and qualitative case studies across multiple datasets. The ablation study effectively isolates the contributions of kernel regression supervision and adaptive kernels. However, generalization beyond text-to-image tasks is not tested. The neighborhood preservation trade-off is acknowledged but not deeply analyzed. While interactive features (zoom, overlay) are highlighted, no user studies or response-time benchmarks are provided. A broader set of cross-modal datasets and human-in-the-loop evaluations would further validate AKRMap’s applicability. Overall, the analysis is strong but could be more comprehensive.

**Methods And Evaluation Criteria:**

The proposed AKRMap method aligns well with the goal of trustworthy cross-modal metric visualization. The use of kernel regression supervision and an adaptive generalized kernel is well-motivated. The evaluation criteria—mapping accuracy (MAE, RMSE, MAPE) on cross-modal metrics (CLIPScore, HPSv2, PickScore, Aesthetic Score)—are appropriate. The HPD dataset and real-world applications (T2I evaluation, model comparison, fine-tuning analysis) provide strong validation. However, broader benchmarks (e.g., video-text, speech-text datasets) would strengthen generalization claims. Neighborhood preservation is evaluated but could be further analyzed in trade-offs with metric accuracy.

**Other Comments Or Suggestions:**

Minor grammatical issues in some sections (e.g., "contour estimation errors back into the projection process" could be clearer).

**Other Strengths And Weaknesses:**

Strengths:
1. Introduces adaptive kernel regression for metric-aware cross-modal visualization, a novel extension to DR methods.
2. Well-structured, with strong visual explanations and quantitative comparisons.

Weaknesses:
1. Only evaluated on text-to-image embeddings, limiting applicability to other cross-modal tasks.
2. Lower than t-SNE/UMAP, with limited discussion on trade-offs.
3. No empirical validation of interactive visualization usability (e.g., response time, interpretability).

**Questions For Authors:**

1. Have you tested AKRMap on other cross-modal tasks (e.g., video-text, speech-text embeddings)? If not, how do you anticipate it would perform in those settings?
2. Given Table 2 shows lower neighborhood trustworthiness than t-SNE/UMAP, have you considered a weighted trade-off mechanism between metric mapping and local structure preservation?
3. Have you conducted any human evaluation to measure the usability of AKRMap’s interactive zoom/overlay features?
4. Does the adaptive kernel regression method have convergence guarantees? Could certain assumptions ensure stability in the projection process?

**Relation To Broader Scientific Literature:**

The paper builds on prior dimensionality reduction (DR) methods like t-SNE, UMAP, and autoencoders, but innovates by integrating adaptive kernel regression for cross-modal metric visualization. It extends supervised t-SNE (Gisbrecht et al., 2015) and parametric UMAP (Sainburg et al., 2021) by incorporating a learnable adaptive kernel. The work is relevant to cross-modal embedding evaluation (CLIPScore, HPSv2) and aligns with efforts in trustworthy AI visualization. Compared to Neuro-Visualizer (Elhamod & Karpatne, 2024), AKRMap achieves better metric fidelity. However, broader validation on other multi-modal tasks (e.g., speech-text embeddings) would strengthen its impact.

**Theoretical Claims:**

The paper does not present formal theoretical proofs but introduces kernel regression-based projection and an adaptive generalized kernel as key theoretical contributions. The formulation of Nadaraya-Watson kernel regression and the adaptive kernel function is mathematically sound. However, the claim that AKRMap optimally balances metric mapping accuracy and neighborhood preservation lacks formal theoretical justification. Additionally, while the adaptive kernel function is empirically validated, a deeper theoretical analysis of its convergence properties or generalization bounds would strengthen its foundation. No significant errors were found, but more rigorous theoretical guarantees would enhance credibility.

---

> ### Author Rebuttal · Authors · 2025-03-31
>
> Thank you very much for your great review helping us improve the work. We respond to your questions below and provide additional figures and tables in the link: https://docs.google.com/document/d/e/2PACX-1vQDqGw_hxxTv_CQbFS_zdI5LtgIE6HTfAYl7uBRgpczNScFBuja6pwbuYrV-76D5CleVrujYDH0O1yP/pub
>
> ## Q1 Have you tested AKRMap on other cross-modal tasks (e.g., video-text, speech-text embeddings)?
>
> A1: Thank you for the suggestion. We have added quantitative experiments on text-audio, image-audio, and text-video embeddings, as well as a qualitative example on image-3D embeddings, demonstrating that our method can generalize to different modalities and tasks. The results are presented in Fig. 1, 2 and Tab. 1, 2, 3 in the link. Please see our response to R1 (m4Lr) Q2 and Q3 for the detailed setup.
>
>
> ## Q2 Given Table 2 shows lower neighborhood trustworthiness than t-SNE/UMAP, have you considered a weighted trade-off mechanism between metric mapping and local structure preservation?
>
> A2: We show the manual adjustment results in response to R3 (63Bw) Q4. Following your suggestion, we test a simple weight balancing technique on the HPSv2 score, where we use a sigmoid function centered at a threshold $\mu$ to adjust the weight of the KL loss:
>  $w(x, \mu) = \sigma(k (x - \mu)) =\frac{1}{1 + e^{-k (x - \mu)}}$,
> $L_1=\lambda MSE_r+ w(KL, \mu) \cdot KL$.
>  Under this setting we find that we can further decrease the mapping error from 0.8 to 0.5. We also test a similar thresholding by $\mu_1$ on the MSE, which can increase the trustworthiness:
>
> ### Table 1: weight balancing result:
> ||mae|mape|rmse|trust|
> |-|-|-|-|-|
> |original|0.811|3.194|1.120|0.828|
> |$\mu=2$|0.512|2.032|0.746|0.805|
> |$\mu_1=0.6$|1.012|1.999|1.348|0.843|
>
> ## Q3 Have you conducted any human evaluation to measure the usability of AKRMap’s interactive zoom/overlay features or response time?
>
> A3: Thank you. We have added a user evaluation with 12 participants for the interactive features, as detailed in Fig. 3 and Tab. 4 in the link. We also record response time on colab notebook. For each zoom interaction on a map, the extra time cost by our dynamic update of contour and sampled points is between 0.0063 and 0.0170 seconds, while the total response time is roughly under 1 second, but it is due to plotly’s innate time cost.
>
> ## Q4 Does the adaptive kernel regression method have convergence guarantees? Could certain assumptions ensure stability in the projection process?
>
> A4: We have attempted to analyze theoretical convergence guarantees for our adaptive generalized kernel, which hold under the following assumptions:
> 1. Parameters must satisfy $\alpha > 0$ and $\beta \geq 1$, ensuring the generalized kernel function $K(x)=(1 + \alpha|x|^{2\beta})^{-1}$ is integrable over $R^2$.
> 2. Certain smoothness conditions (e.g., Lipschitz continuity) on the 2D regression target function $s(x)$ help control the bias term.
> 3. Parameter $\alpha$ should increase appropriately with sample size $N$, analogous to bandwidth reduction in classical kernel regression.
>
> In practice, we enforce non-negative parameters $\alpha, \beta$ by squaring reparameterization, and optimize them via backpropagation. Empirical experiments show successful convergence of learned parameters (e.g., HPSv2: (68.57, 1.61); PickScore: (59.13, 1.35); CLIPScore: (104.70, 3.18); Aesthetic Score: (74.95, 1.11)), validating conditions 1 and 3. The continuous mapping between the embeddings and the metrics like CLIPScore can guarantee condition 2, although specific bounds need more complex derivation.
> Regarding projection stability, our kernel regression loss can be regarded as a regularization term that enforces the global structure of the projection, as it explicitly pushes the projected sample points to regress a global metric distribution.
> We will provide more detailed discussion of the theoretical aspects in updated appendix.
>
> ## Q5 Minor grammatical issues in some sections (e.g., "contour estimation errors back into the projection process" could be clearer).
>
> A5: Thank you. We will fix the issue in the updated version.
>
> ## Q6 Essential references not discussed.
>
> A6: Thank you. We will add discussion of your suggested references in the updated version.
>
> ## Q7 However, the claim that AKRMap optimally balances metric mapping accuracy and neighborhood preservation lacks formal theoretical justification:
> A7: Thank you. We did not intend to claim that AKRMap achieves optimal balance, as we mainly focus on mapping accuracy while the neighborhood only serves as a secondary constraint. In addition, we believe that users can make findings under different trade-off parameters (like tuning perplexity for t-SNE). We will revise our writing to tone down our previous misleading claim. But we also think that your question is very inspiring as how we can define what is the optimal balance between trade-off objectives in all supervised dimensionality reduction methods. We hope to dig deeper into this question in future research.

---

### Official Review · Reviewer_m4Lr · 2025-03-14

**Overall Recommendation:** 4

**Summary:**

The paper introduces a new framework for dimensionality reduction and visualization of cross-modal embeddings. The method incorporates adaptive kernel regression to more accurately capture metric distributions in the projection space. It jointly optimizes a projection network and kernel function, enabling more reliable scatterplots and contour maps with interactive features like zooming and overlaying. Experimental results demonstrate that the proposed method outperforms existing approaches, enhancing the trustworthiness of multi-modal model evaluations, particularly in text-to-image comparisons and human preference assessments.

**Claims And Evidence:**

The claims made in the paper are supported by evidence from both quantitative results and visualizations, demonstrating that the proposed method outperforms other dimensionality reduction techniques. However, the results in Table 1 lack standard deviations and rely on a single dataset (HPD). Additionally, while the visual experiments provide evidence of AKRMap’s ability to handle large datasets, effectively compare model performance, and visualize the impact of fine-tuning, it would be useful to understand how AKRMap generalizes to other cross-modal tasks beyond T2I.

**Essential References Not Discussed:**

I am unaware of any essential references missing from the paper.

**Experimental Designs Or Analyses:**

The method is compared against different existing baselines, evaluated using different metrics. However, the quantitative results lack standard deviations and rely solely on the HPD dataset. Overall, the experimental design is generally sound, but it would benefit from including statistical measures of error and evaluating the method on a broader range of datasets.

**Methods And Evaluation Criteria:**

The proposed methods and evaluation criteria are both theoretically sound and practically relevant for the cross-modal embedding problem in text-to-image generation. However, while the HPD dataset provides a comprehensive evaluation, including experiments on additional datasets would further demonstrate the method's reliability and robustness across diverse embedding scenarios.

**Other Comments Or Suggestions:**

* It would be beneficial to provide the code for AKRMap, either by uploading the files or via [https://anonymous.4open.science/](https://anonymous.4open.science/).
* As mentioned previously, incorporating standard deviations or other statistical error measures in the experimental results would strengthen the paper's claims by providing a more comprehensive view of the performance and reliability of the proposed method.

**Other Strengths And Weaknesses:**

The paper is well-written and easy to follow. The images are clear, and the captions are self-explanatory. The idea is novel, with clear contributions to the field. However, a broader experimental evaluation could strengthen the paper.

**Questions For Authors:**

* Q1: Do the results in Table 1 refer to only out-of-sample points, or do they include both in-sample and out-of-sample points?
* Q2: Could the authors provide quantitative experiments on additional datasets beyond the HPD dataset?
* Q3: Do the authors think the method would work on other modalities beyond text-to-image? Could the authors elaborate on this?

**Relation To Broader Scientific Literature:**

The proposed method builds upon dimensionality reduction, kernel regression, and cross-modal evaluation research, addressing gaps in metric-preserving visualizations. It proposes a novel method for cross-modal embeddings, specifically optimizing dimensionality reduction for cross-modal metrics. Unlike traditional methods, it is designed to enhance metric preservation and interactivity, offering a more trustworthy and interactive visualization approach for large-scale multi-modal datasets.

**Theoretical Claims:**

There are no formal proofs in this paper, but the approach is grounded in empirical design choices with inspiration from prior studies.

---

> ### Author Rebuttal · Authors · 2025-03-31
>
> Thank you very much for your positive feedback and constructive comments that help us improve our work. Below, we answer your specific questions, with two additional figures provided at the link:https://docs.google.com/document/d/e/2PACX-1vR8toOEJ8KrADdAzAXJbIMazphKXlKq4K4oqwDxAvvpLzqPu0jufg7ns67Argf36yCDXGt34dL85eZP/pub
>
> ## Q1: Do the results in Table 1 refer to only out-of-sample points, or do they include both in-sample and out-of-sample points?
> A1: The results in Table 1 refer to only out-of-sample points, while the in-sample points results are presented in Table 3 in appendix B.
>
> ## Q2: Could the authors provide quantitative experiments on additional datasets beyond the HPD dataset?
> A2: We have added experiments on three different types of embeddings on two additional datasets: ImageBind audio-text embeddings and audio-image embeddings on Flickr 8k Audio Caption dataset, as well as CLIP text-video embeddings on MSR-VTT datasets, showing that our method can improve upon traditional DR in test set mapping accuracy (Please refer to Table 1-3 below). The results are as follows:
> ### Table 1. ImageBind text-audio embeddings mapping accuracy on Flickr 8k Audio Caption Corpus.
> ||PCA|t-SNE|UMAP|AKRMap|
> |-|-|-|-|-|
> |mae|3.7194|3.5992|3.6801|1.8629|
> |mape|21.1113|20.3858|20.8498|10.1821|
> |rmse|4.7175|4.5607|4.6541|2.4841|
> ### Table 2. ImageBind image-audio embeddings mapping accuracy on Flickr 8k Audio Caption Corpus.
> ||PCA|t-SNE|UMAP|AKRMap|
> |-|-|-|-|-|
> |mae|4.1239|3.9576|3.8811|1.9425|
> |mape|19.1321|18.3808|18.1024|8.9521|
> |rmse|5.1737|5.0254|4.8633|2.6425|
> ### Table 3. CLIP text-video embeddings mapping accuracy on MSR-VTT.
> ||PCA|t-SNE|UMAP|AKRMap|
> |-|-|-|-|-|
> |mae|1.4908|1.4201|1.4562|0.6763|
> |mape|8.3080|7.9589|8.1636|3.4832|
> |rmse|2.1231|2.0216|2.0723|0.9578|
>
> ## Q3: Do the authors think the method would work on other modalities beyond text-to-image? Could the authors elaborate on this?
> A3: Our method works effectively across various modalities and shows potential for further extension depending on the complexity of embeddings. As mentioned in A2, we've performed additional experiments on text-audio, image-audio, and text-video embeddings with positive results (please refer to Tables 1, 2, and 3 in Q2 for quantitative results), and we also provide visualization of dimensionality reduction results for text-to-video in Figure 1 in our provided link. In addition, we've also successfully applied our approach to image-to-3D tasks, where we visualized and compared two video-diffusion-based 3D generation models on 360-degree videos of generated 3D assets from the GSO dataset (Figure 2 in our provided link). Some complex cases such as 3D and video generation involve multimodal embeddings (e.g., CLIP) for evaluation. Even though these tasks require embeddings from multiple views of a single instance, we can average these embeddings before projection, as the linear operation in averaging is associative with the dot product operation used in computing CLIPScore.
>
> ## Q4: The results in Table 1 lack standard deviations.
> A4: Thank you for the suggestion. We have rerun most of regression tests 10 times in Table 1 (excluding the auto-encoder method that takes too long to run and they are clearly worse than other methods in previous experiments) to obtain the standard deviation:
> ### Table 4. Standard deviation for HPSv2
> ||PCA|t-SNE|UMAP|w/o KR|w/o GK|AKRMap|
> |-|-|-|-|-|-|-|
> |mae| 0.0073 | 0.0063 | 0.0079 | 0.0054 | 0.0033 | 0.0058|
> |mape| 0.030| 0.026 | 0.035 | 0.023 | 0.015 | 0.023|
> |rmse| 0.010 | 0.0089 | 0.012 | 0.0060 | 0.0047| 0.0085|
> ### Table 5. Standard deviation for CLIPScore
> ||PCA|t-SNE|UMAP|w/o KR|w/o GK|AKRMap|
> |-|-|-|-|-|-|-|
> |mae| 0.016 | 0.016  | 0.019 | 0.011 | 0.0051 | 0.0042 |
> |mape| 0.036 | 0.041 | 0.080 | 0.026 | 0.028 | 0.018 |
> |rmse| 0.013 | 0.012 | 0.037 | 0.010 | 0.011 | 0.0046 |
>
> ### Table 6. Standard deviation for PickScore
> ||PCA|t-SNE|UMAP|w/o KR|w/o GK|AKRMap|
> |-|-|-|-|-|-|-|
> |mae| 0.0061 | 0.0053 | 0.0080 | 0.0024 | 0.0029 | 0.0032 |
> |mape| 0.021 | 0.024 | 0.031 | 0.015 | 0.016 | 0.0086 |
> |rmse| 0.0055 | 0.0057 | 0.0069 | 0.0031 | 0.0037 | 0.0042 |
>
> ### Table 7. Standard deviation for Aesthetic Score
> ||PCA|t-SNE|UMAP|w/o KR|w/o GK|AKRMap|
> |-|-|-|-|-|-|-|
> |mae| 0.0028 | 0.0047 | 0.0027 | 0.0019 | 0.0033 | 0.0026|
> |mape| 0.077 | 0.12 |0.070 | 0.048 | 0.064 | 0.071|
> |rmse| 0.0037 | 0.0059 | 0.0033 | 0.0021 | 0.0030 | 0.0032 |
>
> ## Q5: It would be beneficial to provide the code for AKRMap, either by uploading the files or via https://anonymous.4open.science.
> A5: Thank you. To facilitate convenient test of the code which requires large precomputed embedding npy files exceeding github upload limit, we have uploaded the code files to an anonymous osf: https://osf.io/7mwv3/?view_only=18869f3a1cf542b4bf2f39db4e19abd7. The link may take a while to load and it is better to paste it on another tab. You can find the files under the Files tab. We will open source the code on github after the review period.

---

> > ### Comment · Reviewer_m4Lr · 2025-04-04
> >
> > I thank the authors for their comprehensive response, particularly for including additional experiments that demonstrate the effectiveness of the proposed method across different modalities. I will maintain my score of 4 and recommend acceptance.

---

> > > ### Author Response · Authors · 2025-04-06
> > >
> > > Thank you very much for your acknowledgement and feedback. Your fantastic review and positive comments are inspirational for a student pursuing interesting and meaningful research on human-AI interaction techniques, which I believe holds a unique position with increasingly powerful models and exponentially growing volume of generated data coming into humans' every day life.

---

### Decision · Program_Chairs · 2025-05-01

**Decision:**

Accept (poster)

**Comment:**

This paper introduces a new approach to visualizing cross-modal data. While the method is theoretically underdeveloped, it outperforms other methods and the paper includes good experimental studies and interesting applications. The reviewers all lean toward acceptance. Thus I recommend that the paper be accepted.